

# CALIPSO IIR Version 2 Level 1b calibrated radiances: analysis and reduction of residual biases in the Northern Hemisphere

Anne Garnier[1,2], Thierry Trémas[3], Jacques Pelon[4], Kam-Pui Lee[1,2], Delphine Nobileau[5], Lydwine Gross-Colzy[5], Nicolas Pascal[6], Pascale Ferrage[3], Noëlle A. Scott[7]

[1]Science Systems and Applications, Inc., Hampton, VA 23666, USA
[2]NASA Langley Research Center, Hampton, VA 23681, USA
[3]Centre National d'Études Spatiales, Toulouse, 31401, France
[4]Laboratoire Atmosphères, Milieux, Observations Spatiales, UPMC-UVSQ-CNRS, Paris, 75252, France
[5]Cagpemini Technology Services, Toulouse, 31086, France
[6]Hygeos, AERIS/ICARE Data and Services Center, Lille, 59650, France
[7]Laboratoire de Météorologie Dynamique, Ecole Polytechnique-CNRS, Palaiseau, 91128, France

*Correspondence to*: Anne Garnier (anne.garnier@latmos.ipsl.fr)

**Abstract.** A new Version 2 of the Level 1b calibrated radiances of the Imaging Infrared Radiometer (IIR) onboard the Cloud-Aerosol Lidar and Infrared Satellite Observation (CALIPSO) satellite has been released recently. This new version incorporates corrections of small but systematic seasonal calibration biases previously revealed in Version 1 data products mostly north of 30° N. These biases, of different amplitudes in the three IIR channels 8.65 µm (IIR1), 10.6 µm (IIR2), and 12.05 µm (IIR3), were made apparent by a striping effect in images of IIR inter-channel brightness temperature differences (BTDs) and through seasonal warm biases of nighttime IIR brightness temperatures in the 30° N-60° N latitude range. The latter were highlighted through observed and simulated comparisons with similar channels of the Moderate Resolution Imaging Spectroradiometer (MODIS) onboard the Aqua spacecraft. To characterize the calibration biases affecting Version 1 data, a semi-empirical approach is developed, which is based on the in-depth analysis of the IIR internal calibration procedure in conjunction with observations such as statistical comparisons with similar MODIS/Aqua channels. Two types of calibration biases are revealed: an equalization bias affecting part of the individual IIR images and a global bias affecting the radiometric level of each image. These biases are observed only when the temperature of the instrument increases and they are found to be functions of elapsed time since night-to-day transition, regardless of the season. Correction coefficients of Version 1 radiances could thus be defined and implemented in the Version 2 code. As a result, the striping effect seen in Version 1 is significantly attenuated in Version 2. Systematic discrepancies between nighttime and daytime IIR-MODIS BTDs in the 30° N-60° N latitude range in summer are reduced from 0.2 K in Version 1 to 0.1 K in Version 2 for IIR1-MODIS29. For IIR2-MODIS31 and IIR3-MODIS32, they are reduced from 0.4 K to close to zero, except for IIR3-MODIS32 in June where the night-minus-day difference is around -0.1 K.





# 1 Introduction

Since 2006, the Cloud-Aerosol Lidar and Infrared Satellite Observation (CALIPSO) satellite has been providing a quasi-3D description of the atmosphere with vertically resolved cloud and aerosols properties from the Cloud and Aerosol Lidar with Orthogonal Polarization (CALIOP), complemented by passive observations in the thermal infrared atmospheric window from

the Imaging Infrared radiometer (IIR) and in the visible spectral range from the Wide Field of view Camera (WFC) (Winker et al., 2010). CALIPSO, which is part of the A-Train constellation (Stephens et al., 2002), follows a Sun-synchronous orbit at an altitude of 705 km with an inclination of 98.2°. The three instruments are assembled in a staring and near nadir looking configuration. The IIR includes three medium resolution channels at 8.65 µm (IIR1), 10.6 µm (IIR2), and 12.05 µm (IIR3) with bandwidths of 0.85 µm, 0.6 µm and 1 µm, respectively. The IIR calibrated radiances are reported in the IIR Level 1b

product (Vaughan et al., 2017), where they are registered on a 1-km resolution grid centered on the CALIOP ground track, with a 69-km swath. The calibrated radiances are often expressed in terms of equivalent brightness temperatures computed using the relevant instrument spectral response functions.

CALIPSO provides simultaneous and collocated retrievals of cirrus optical depths in the visible from CALIOP (Young and Vaughan, 2009; Young et al., 2013) and in the thermal infrared from IIR, with different sources of uncertainties, thereby

allowing detailed mutual assessment (Garnier et al., 2015). IIR also provides ice crystal effective diameters, which are derived from two microphysical indices defined as the ratios of the effective infrared optical depths in the two pairs of channels 12.05-10.6 µm and 12.05-08.65 µm (Garnier et al., 2013). The accuracy of IIR optical depth retrievals and of subsequent microphysical indices depends in part on the accuracy of the calibrated radiances. For instance, for oceanic cirrus clouds of extinction optical depths of 0.3, 0.5, and 2.5, an uncertainty of 0.3 K in the measured equivalent brightness temperature induces

typical relative uncertainties of 8 %, 5 %, and 2 % in the retrieved optical depth (Garnier et al., 2015). Inter-channel calibration biases induce errors in the microphysical indices and therefore can affect the microphysical retrievals.

Until recently, the sole version of the IIR level 1b product has been Version 1, with no changes to the calibration procedure since launch. Nevertheless, a striping effect was noticed soon after launch in images of inter-channel brightness temperature differences (BTDs) over homogeneous scenes (Trémas, 2006; Scott, 2009). This striping effect occurs in the Northern

Hemisphere, typically north of 30° N with a spatial periodicity of about 50 km. In parallel, in order to assess Version 1 calibration stability and accuracy, Version 1 calibrated radiances have been monitored since the beginning of the CALIPSO mission through two concomitant approaches based on simulated and observed comparisons with similar channels of the Moderate Resolution Imaging Spectroradiometer (MODIS) onboard the Aqua spacecraft (Scott, 2009; Garnier et al., 2017 (henceforth G17); Scott et al., 2017). Time series analyses were carried out by averaging the individual observations with

latitudinal resolutions of several tens of degrees. Excellent stability and accuracy of the Version 1 IIR calibrated radiances were found (G17), well within the specified accuracy of 1K in all channels. However, unexplained seasonal night/day differences of up to 0.4 K in June and July were made evident in the 30° N-60° N latitude band, with remarkable repeatability



since launch. Analyses revealed that this phenomenon originates from IIR and is due to warm biases in Version 1 nighttime IIR brightness temperatures in this latitude range.

Both the striping effect and the warm biases in the nighttime IIR calibrated radiances were seen typically north of 30° N. These two issues have motivated a detailed examination of the IIR internal calibration procedure and the search for possible sources

of biases. The study was carried out by coupling instrumental standpoint, based on the analysis of internal calibration data available to the Centre National d'Études Spatiales (CNES) IIR Technical Expertise Center, and observations from the IIR Level 1b product. Calibration biases could then be characterized, and corrections were established using a semi-empirical approach. These corrections have been implemented in the new Version 2 of the IIR Level 1b products which was released in July 2017.

This paper presents the subsequent steps in the development of the Version 2 IIR Level 1b products. After a brief description of the IIR Level 1b product in Sect. 2, the observations that highlighted issues in Version 1 data products and motivated this work are presented in Sect. 3. Findings from the synergetic analysis of the internal calibration and of the observations are developed in Sect. 4. Based on these findings, calibration corrections could be established by following the rationale presented in Sect. 5. Results obtained with Version 2 and improvements with respect to Version 1 are shown and discussed in Sect. 6,

followed by Sect. 7 which concludes the paper.

## 2 IIR Level 1b product

### 2.1 IIR instrument

The IIR instrument (Corlay et al., 2000) includes three medium-resolution channels and one unique sensor: an uncooled microbolometer array (U3000A) manufactured by the Boeing company. An individual measurement is an image composed of

20 64 rows x 64 columns. The rows are oriented cross-satellite track and the columns are parallel to the satellite track. The size of an individual pixel is 1 km$^2$, so that each individual image covers 64 km x 64 km. IIR includes three filters arranged on a filter wheel for sequential acquisition in the three channels. The instrument is regularly calibrated using images from cold (about 4 K) deep space views and from a warm blackbody source of measured temperature, on the order of 25°C.

### 2.2 IIR data acquisition

The IIR acquisition timing is organized around successive cycles described in Table 1. Each cycle is composed of five sequences. Each of these five sequences includes three successive calibration images -one per channel, followed by three successive Earth view images in each channel. In each cycle, the calibration images are blackbody (BB) views in the first sequence, and are deep space (DS) views in the other four sequences. The total duration of a cycle is 40.92 s, i.e. 8.184 s per sequence. The elapsed time between two successive Earth views in a given channel is 8.184s, during which the satellite has

moved forward by about 55 km, so that two successive Earth view images always overlap.


The duration of a full orbit corresponds to about 145 cycles. In this paper, IIR acquisition cycles are counted from cycle #0 defined as the first cycle after night-to-day transition. Thus, cycle number is a measure of elapsed time since night-to-day transition. The notion of night and day follows the definition chosen for the CALIOP products, with the daytime portion of an orbit corresponding to solar elevation angles at Earth surface larger than -5° (Hunt et al., 2009). The relationship between IIR

cycle number and latitude is shown in Fig. 3 for four months representative of the four seasons. Cycle #0 is located near the pole in the Southern Hemisphere. The ascending (descending) portions of the orbits are where latitude increases (decreases) as cycle number increases. The relationship between IIR cycle number and latitude is season-dependent, because it is a function of the season-dependent location of the night-to-day transition.

### 2.3 IIR Level 1 processing

The IIR Level 1 processing includes two major steps. First, each individual 64x64 Earth view image is calibrated following the internal procedure presented below in this section. After calibration in each channel, the individual 64x64 calibrated Earth view images are projected using a bi-cubic interpolation onto a unique geo-located grid at sea level centered on the CALIOP lidar track, with 1 km$^2$ pixel resolution and a 69-km swath. After projection, the rows and columns from the individual images are nearly cross-lidar track and parallel to the track, respectively.

The internal calibration consists in calibrating each pixel of each individual Earth view image by using surrounding DS and BB views (see Table 1). For each channel, and for each pixel in a row (i) and in a column (j) of an individual 64x64 Earth view image in a sequence s, the raw digital counts $X_E(i,j,s)$ are calibrated as follows. First, $X_E(i,j,s)$ is corrected for the offset measured during surrounding DS views. Then, the corrected raw digital counts are converted into calibrated radiances through the gain, $\overline{G}(i,j,s)$. Thus, the calibrated radiance R(i,j,s) in units of $W.m^{-2}.sr^{-1}.\mu m^{-1}$ is written as:

$$R(i,j,s) = \left(X_E(i,j,s) - offset\right) \times \frac{1}{\overline{G}(i,j,s)} \qquad (1)$$

The offset is obtained by averaging digital counts from the eight or nine surrounding DS views. The gain $\overline{G}(i,j,s)$ is obtained by averaging four individual gains associated to the four BB views surrounding the sequence s. An individual gain G(i,j,c) derived from the BB view in a cycle c is computed as:

$$G(i,j,c) = \frac{R_{BB}(c)}{\left(X_{BB}(i,j,c) - offset_{BB}\right)} \qquad (2)$$

where $R_{BB}(c)$ is the blackbody radiance associated with its measured temperature $T_{BB}(c)$, $X_{BB}(i,j,c)$ are the digital counts in the BB view, and $offset_{BB}$ is the offset correction obtained by averaging the digital counts from the eight closest DS views. The number of DS views used to compute the offsets and the number of individual gains used to compute the gain $\overline{G}(i,j,s)$ result from a compromise between noise reduction and calibration accuracy. They were established before launch and confirmed during the in-flight performances assessment (Trémas, 2006).



## 3 Motivation for a change

The need to improve the Version 1 IIR Level 1b data product was motivated by two different types of issues, independently highlighted after the analysis of numerous years of Version 1 data. The first issue was the striping effect, which was detected through the visual inspection of IIR browse images. The second issue, the presence of biases at 30° N-60° N, was made evident

after careful statistical comparisons with MODIS/Aqua. Even though these two issues seemed unrelated at first glance, this study demonstrates that they are not and that both can be corrected simultaneously.

### 3.1 Striping effect

The striping effect mentioned earlier refers to the presence of stripes in images of IIR inter-channel BTDs. The striping effect is best seen over scenes that are sufficiently homogeneous in terms of inter-channel BTDs. Thus, it is best seen over cloud-

free scenes since clouds may induce additional variable inter-channel BTDs (Inoue, 1985; Ackerman et al., 1990). Water surface is also more favourable, because surface emissivity is more homogeneous than over land. An illustration is given in Fig. 1, which shows IIR1-IIR3 and IIR2-IIR3 inter-channel BTDs over the IIR 69-km swath in the nighttime descending portion of an orbit between 46° N and 43° N over water surface in June 2012. The CALIOP lidar measurements, at the center of the IIR swath, indicate a cloud-free scene with aerosol layers at low altitude. Regularly spaced darker stripes are clearly

seen for both BTDs, with a spatial periodicity of about 0.5° in latitude. The amplitude of this striping effect, unambiguously an artifact, varies with latitude and season. and is seen typically north of 30° N. Figure 1 shows a worst-case example deliberately chosen for illustration purposes. In this case, the negative anomaly in the inter-channel BTDs associated to the darker stripes is about -0.5 K. The periodicity of about 0.5° in latitude represents about 50 km along the track, which corresponds to about one individual image, and therefore strongly suggests an artifact related to the IIR acquisition. The stripes

are quasi-perpendicular to the CALIOP track and likely follow the columns of the individual images. This suggests that the striping effect is mostly related to some rows of the individual images, as will be discussed in Sect. 4.

### 3.2 Seasonal IIR-MODIS night/day differences at 30° N-60° N

Assessing the IIR radiances required external comparison with other instruments. For this purpose, comparisons with MODIS/Aqua have been carried out since launch by following two complementary methods: (i) the "relative" instrument-to-

instrument inter-comparison approach, and (ii) the "stand alone" simulations-to-observations comparison approach (Scott, 2009; G17). For both approaches, IIR channels IIR1, IIR2, and IIR3 were paired with Collection 5 (C5) MODIS/Aqua channels 29, 31, and 32, respectively. Collocated IIR and MODIS observations are from the "REMAP'' product available at the AERIS/ICARE Data and Services Center (http://www.icare.univ-lille1.fr). IIR and MODIS pixel sizes are similar (1 km²) and IIR and MODIS/Aqua are both in the A-Train, so a lot of collocated quasi-simultaneous observations are available, with

MODIS viewing angles for pixels collocated with IIR decreasing from 20° at the equator to a few degrees near the poles. Pre-launch simulations showed that the expected difference between the brightness temperatures in the IIR and MODIS paired



channels is -1K to +0.3 K (with standard deviation from 0.33 to 0.02K) depending on air mass type and viewing angle. Using the relative approach, time series of daily averaged IIR-MODIS BTDs over oceans have been analyzed since launch. Figure 2 shows such time series in the 30° N-60° N latitude band for brightness temperatures ranging between 280 K and 290 K, by separating nighttime (blue) and daytime (red) IIR-MODIS BTDs. As mentioned in the introduction, unexplained seasonal night/day differences are observed, with remarkable repeatability since launch, while such differences are not seen in southward latitude ranges (G17). The worst cases are in June and July, during which the mean nighttime BTDs are larger than the daytime BTDs by up to 0.2 K for IIR1 and 0.4 K for IIR2 and IIR3. These inter-comparisons have been further assessed using the stand-alone approach through time series since launch of [simulated-observed] BTDs, also called residuals, for each IIR and MODIS channel in clear sky conditions over oceans (G17). The simulations were performed using the 4A/OP radiative transfer model and the Gestion et Étude des Informations Spectroscopiques Atmosphériques (GEISA) spectroscopic database (http://ara.lmd.polytechnique.fr), with atmospheric and surface inputs from 3h/5km collocated ERA-Interim products (Dee et al., 2011). The clear sky mask was based on co-aligned observations from CALIOP and IIR, further extended to the IIR 69-km swath. The stand-alone approach showed that the slow decrease over time of both nighttime and daytime IIR1-MODIS29 BTDs as seen in Fig. 2 is due to a drift in MODIS29 C5 measurements. Relevant to the present work, it also revealed that the night/day differences originated from IIR and were due to warm biases in Version 1 nighttime IIR brightness temperatures in the 30° N-60° N latitude range (G17).

## 4 Version 1 internal calibration analysis

The first step of the study is to search the origin of the biases found in Version 1 (Sect. 3). To that end, the IIR internal calibration procedure (Sect. 2) is now examined in details. The striping effect and comparisons with MODIS are then revisited according to information from the internal calibration analysis. Because the issues were detected in the Northern Hemisphere at season-dependent latitudes, the internal calibration is analyzed as a function of IIR acquisition cycle number, which is counted from elapsed time since the night-to-day transition (Fig. 3).

### 4.1 Evidence of IIR cycle-dependent sound and flawed rows

The internal calibration has been analyzed by looking at the behavior of the calibration DS and BB views as functions of IIR cycle number. Digital counts in both the DS and BB views exhibit variations along the orbit that are correlated with the changing temperature of the instrument, which is expected. For the DS views, similar variations are seen for all the pixels of an individual image. However, for the BB views, the time variation of the digital counts of the measured signal $X_{BB}(i,j,c)$ is not always the same for all the rows (i), which is still true after correcting $X_{BB}(i,j,c)$ for $offset_{BB}$ determined from surrounding DS views (cf Eq. (2)). However, for a given row, all the columns (j) behave similarly. As an illustration, Fig. 4 (top) shows the mean values of the $X_{BB}(i,c)$-$offset_{BB}$ digital counts vs. IIR cycle number during June 2012 in each IIR channel. Each panel shows 64 curves for each of the 64 rows, after averaging the 64 columns associated with a given row in order to reduce the



noise. The resulting mean gains G(i,c) derived from Eq. (2) are also plotted (Fig. 4, bottom). The different gains in IIR1, IIR2, and IIR3 reflect the different optical efficiencies through their respective filters. Before cycle #36 and after cycle #85, the time variation of the digital counts (top) is similar for all the rows, and follows the time variation of the blackbody radiance $R_{BB}(c)$, yielding fairly stable gains for all the rows (bottom). However, differences are clearly seen between cycles #36 and #85. Rows

0 to 31 and 60 to 63, in purple, continue to have time variations similar to $R_{BB}(c)$, with a slow decrease of the gains up to cycle #50 followed by a slow increase up to cycle #85. In contrast, for the other rows plotted in green, $X_{BB}(i,c)$-offset$_{BB}$ increases rapidly after cycle #36 before dropping abruptly at cycle #85 to return to "stable" values. As a result, the gain increases more rapidly for the green rows than for the purple rows. These observations are applicable to the three channels, but with different relative amplitudes.

The distinct behavior of some rows between cycles #36 and #85 occurs when the temperature of the instrument increases, suggesting a thermally induced effect. Remarkable repeatability of this phenomenon is observed since the CALIPSO launch, regardless of the season. The repeatability from one season to another is explained by the fact that the IIR acquisition cycles are counted with respect to the night-to-day transition. Because the latter occurs at season-dependent latitudes, the latitudes corresponding to cycles #36 and #85 also depend on the season (see Fig. 3). In June, cycle #36 occurs during the daytime

ascending portion of the orbit at about 25° N and cycle #85 occurs during the nighttime descending portion at about 33° N. In December, cycle #36 occurs again during the daytime ascending portion of the orbit, but at about 25° S; and cycle #85 occurs at the beginning of the nighttime descending portion of the orbit at about 79° N.

Two families of rows have been defined. The purple rows (0-31 and 60-63) exhibiting similar time variation of $X_{BB}(i,c)$-offset$_{BB}$ and $R_{BB}(c)$ are considered "sound" rows, and the green rows (32-59) are considered "flawed" rows. The procedure

developed for the calibration correction will be based on this classification, as will be discussed in Sect. 5. The correction procedure will also be guided by the observed impact on Version 1 calibrated radiances, which will now be discussed.

**4.2 Impact on Version 1 calibrated radiances**

The striping effect and comparisons with MODIS are now revisited in light of the above, i. e. the evidence of sound and flawed rows of different behavior between IIR acquisition cycles #36 and #85. The comparisons with MODIS are analyzed according

to IIR acquisition cycle, which appears to be a key parameter to relate internal calibration with observations.

As presented in Sect. 2, the geolocated calibrated radiances reported in the Level 1b product are obtained by projecting the individual calibrated images onto a unique grid centered on the CALIPSO lidar track. Thus, identifying the various rows of an image in the Level 1b product is not straightforward. Therefore, a new flag was implemented for this study, which indicates, for each IIR channel, whether a Level 1b pixel originated from a sound or a flawed row.

**4.2.1 Striping effect**

Figure 5 shows the IIR1-IIR3 and IIR2-IIR3 BTDs for the same cloud-free scene over water surface as in Fig. 1, between 46° N and 43° N, for the IIR pixels located along the lidar track, at the center of the swath. Fig. 5 covers about 6 sequences, that is





1.2 times an IIR cycle, around cycle #80 (Fig. 3). Because of the sequential acquisition of the 64x64 images in the three IIR channels, a given pixel in the Level 1b product does not originate from the same rows in the three channels. Using the newly implemented flag, pixels originating from sound rows in both channels (in purple) are distinguished from the other pixels (in green). The regularly spaced negative spikes around -2 K in IIR1-IIR3 and 0-0.5 K in IIR2-IIR3 correspond to the darker

stripes seen over the swath in Fig. 1. Interestingly, they all originate from flawed rows (green). The stripes seen over the swath are due to the fact that the successive rows are oriented nearly cross lidar track and that all the columns of a given row behave similarly. Thus, the impact of the sound and flawed rows on the Version 1 calibration can be assessed from the track pixels. The regularly spaced dark stripes (negative spikes) indicate systematic relative calibration biases between rows. In other words, they indicate intra-image calibration biases.

**4.2.2 Comparisons with MODIS along the CALIOP track**

After the calibration biases demonstrated through comparisons with MODIS paired channels using the relative approach, the comparisons with MODIS were refined in order to integrate information related to the IIR internal calibration. As in the relative approach, the pairs of IIR and MODIS channels considered for the comparisons are IIR1-MODIS29, IIR2-MODIS31, and IIR3-MODIS32. MODIS data is from C5, but using MODIS Collection 6 (Toller et al., 2013) would not change the discussion

(not shown). For simplicity, the calibrated radiances are assessed now using only the track pixels. Cloud-free conditions determined using the CALIOP 5-km cloud layer products (Vaughan et al., 2017) are selected to favor homogeneous scenes and facilitate the reasoning. Figure 6 shows median IIR-MODIS BTDs for the three pairs of channels vs. latitude over oceans. The BTDs were computed vs. IIR cycle number by averaging the pixels of the five consecutive sequences in each cycle (see Table 1), and then plotted vs. latitude using the cycle number-latitude relationships as illustrated in Fig. 3. Daytime and

nighttime portions of the orbits are in solid and dashed lines, respectively. Furthermore, between cycles #36 and #85, IIR-MODIS BTDs are computed separately for IIR pixels originating from sound rows (purple) and from flawed rows (green) using the dedicated flag implemented for this study.

Two main conclusions can be drawn from Fig. 6.

First, we see by comparing the purple and green curves that the BTDs associated to the sound and to the flawed rows

progressively depart from each other starting around 45° N during the daytime ascent until about 35° N in the nighttime descent. The largest differences are 0.3 K for IIR1-MODIS29 and 0.2 K for IIR2-MODIS31, and are smaller than 0.1 K for IIR3-MODIS32. Thus, statistical comparisons with MODIS at a resolution of several individual images show that the sound and the flawed rows are on average not calibrated in a consistent manner. These observations, as well as the intra-image calibration biases made evident by the striping effect (Sect. 4.2.1), can be traced back to the gains of the sound and flawed

rows described in Sect. 4.1, and indicate that an equalization correction is required.

Secondly, an unexpected hysteresis effect is clearly seen in Fig. 6 north of 35° N for both the sound and the flawed rows, which indicates a second type of "global" calibration bias affecting in this case all pixels of an image. Because of this global calibration bias, IIR-MODIS BTDs between 30° N and 60° N are systematically larger at night than during the day for this





June example, which explains the differences seen in Fig. 2 for every summer since launch. Interestingly, the amplitude of the hysteresis effect is smaller for the flawed rows than for the sound rows, for which the amplitude is up to 0.5 K. This is particularly obvious in IIR1-MODIS29, and interestingly IIR1 happens to be the channel for which the calibration bias between the sound and the flawed rows is the largest. Similar conclusions could be drawn from the analysis of several months

representative of other seasons. The global calibration bias with respect to MODIS is synchronized with IIR cycle number and is typically concomitant with the equalization bias.

These findings will guide the definition of a corrected calibration procedure, as presented in Sect. 5.

## 5 Internal calibration corrections

Based on the analyses and observations presented in Sect. 4, a two-step correction procedure has been defined for Version 2.

First, an equalization correction is to be applied to some rows of each image. Subsequently, the radiometric level of each image must be corrected to reduce the systematic global bias with respect to MODIS. Because of the observed repeatability of both issues since launch, the chosen approach was to define correction coefficients synchronized with the IIR cycle number. The intra-image and the global biases are corrected through two series of tables, each series made up of one table for each IIR channel. The gain corrections were defined after extensive analysis of two test months from opposite seasons, namely June

2012 and January 2010. The rationale for the definition of the equalization and global bias corrections is presented in the following Sects. 5.1 and 5.2, and the resulting correction coefficients implemented in Version 2 are shown in Sect. 5.3.

### 5.1 Equalization correction

Because the offset-corrected BB digital counts of the sound rows exhibit time variations similar to $R_{BB}(c)$ (cf Sect. 4.1), the equalization correction is defined by using the behavior of sound rows as a reference. For each flawed row (i=32 to 59), the

20 difference $X_{BB}(i)$-offset$_{BB}$ is required to have the same time variation as the mean $X_{BB}(i)$-offset$_{BB}$ of the reference rows. The chosen references are rows 0 to 20, because among the sound rows, those are the ones that exhibit the closest behavior. Rows 21 to 31 and 60 to 63 still qualify as sound rows because their time variations of $X_{BB}(i)$-offset$_{BB}$ always differ by less than 1.5 % from the reference rows. The same procedure is applied independently to each of the three channels. The two success criteria are attenuation of the striping effect (Figs. 1 and 5) and closer agreement between the IIR-MODIS BTDs of the sound and

25 flawed rows (Fig. 6). Initial attempts to apply the correction between cycles #36 and #85 showed an over-correction. The best compromise for a satisfactory correction of all three IIR channels was obtained by starting the correction at cycle #46. The mean values of $X_{BB}(i)$-offset$_{BB}$ and of the resulting gains $G_1(i,c)$ obtained after equalization correction during June 2012 are shown in Fig. 7. The only differences between Fig. 7 and Fig. 4, which was before correction, are for the flawed rows between cycles #46 and #85. The mean values of $X_{BB}(i)$-offset$_{BB}$ of the flawed rows before correction are added to Fig. 7 using green

dashed lines for visual comparison. After equalization correction, $X_{BB}(i)$-offset$_{BB}$ and the gain of the flawed rows are decreased, which increases their radiances (see Eq. (1)) and therefore their brightness temperatures.





## 5.2 Global bias correction

After equalization correction, the radiances of the sound rows are unchanged while the radiances of the flawed rows are increased. Therefore, the hysteresis effect highlighted in Sect. 4.2.2 and seen in Fig. 6 is still present, but it is now of similar amplitude for the sound and the flawed rows. Unfortunately, no evidence of systematic calibration biases that would explain

this hysteresis effect could be derived from the analysis of the pre-Level 1b data or from the available IIR instrumental data. Therefore, the correction of the systematic calibration biases had to rely on an empirical approach with the goal to reduce the hysteresis effect seen in the IIR-MODIS BTDs.

As noted earlier, before the equalization correction (Fig. 6), the hysteresis effect is the smallest for the flawed rows of IIR1 (top). As seen in Fig. 7 (top, left), $X_{BB}(i)$-offset$_{BB}$ of the flawed rows in IIR1 before equalization correction (dashed green)

between cycle #51 and cycle #85 increases much more rapidly with IIR cycle number (or time) than for the sound rows. This larger increase of $X_{BB}(i)$-offset$_{BB}$ translates into a larger increase of the gain and into a smaller increase of the brightness temperatures, leading to a smaller amplitude of the hysteresis effect for instance around 40° N, which corresponds to cycle #42 in the ascending daytime orbits and to cycle #82 in the nighttime descending orbits (Fig. 3). After equalization correction, $X_{BB}(i)$-offset$_{BB}$ increases quasi-linearly with time between cycles #51 and #85 for all the rows, but based on the above

observations, which could be repeated for other months, it appears that the slope should be steeper. We chose to correct the slope by using the mean increase of $X_{BB}(i)$-offset$_{BB}$ of the flawed rows before equalization correction as a guide. Using this approach, the resulting gain in IIR1 was increased quasi-linearly by about 1 between cycles #51 and #85. A satisfactory correction for the three channels was obtained by applying the same gain changes to IIR2 and IIR3. The resulting gains $G_2(i,c)$ in channels IIR1, IIR2, and IIR3 are shown in Fig. 8 for the month of June 2012. The gains are augmented by the same absolute

value for the three IIR channels, so that the relative change differs from one channel to the other. The fact that the corrected gains are found to increase more rapidly between cycles #51 and #85 than the gains derived after equalization correction (see Fig. 7) suggests the presence of an additional parasitic contribution to the digital counts in the Earth view images. Because the increase is the same in absolute value for the three channels, this parasitic contribution is likely independent of the filters' optical transmission. As far as the blackbody images are concerned, this parasitic contribution is not seen in the sound rows,

and some parasitic contribution seems to be present for the flawed rows, especially in IIR1.

## 5.3 Version 2 correction coefficients

The analysis presented in Sects. 5.1 and 5.2 has been conducted for both the months of June 2012 and January 2010. For both test months, similar results were obtained in terms of relative change of the corrected gains $G_1(i,c)$ and $G_2(i,c)$ with respect to the Version 1 gains $G(i,c)$ as a function of the IIR cycle number. Based on these results, two series of correction coefficients

have been defined for implementation in a new Version 2 of the IIR Level 1 code. For any month of the CALIPSO archive, the calibrated radiances are first computed as in Version 1 ($R_{V1}$), and then corrected using these correction coefficients.





For each IIR channel (k), the first series of coefficients $C_{eq}(i,c,k)$ defines the calibrated radiance equalization correction for each row (i) as a function of the IIR cycle number. For each IIR channel (k), $C_{eq}(i,c,k)$ is the ratio of $G(i,c)$ to $G_1(i,c)$. The second series of coefficients $C_{bias}(i,c,k)$ corrects for the global bias found through comparisons with MODIS and is defined as the ratio of $G_1(i,c)$ to $G_2(i,c)$. Finally, for each pixel in a row (i), the Version 2 calibrated radiances ($R_{V2}$) are computed as:

$$R_{V2}(i,c,k) = \left( C_{eq}(i,c,k) \cdot C_{bias}(i,c,k) \right) \times R_{V1}(i,c,k) \qquad (3)$$

The two series of correction coefficients applied in Version 2 are shown in Fig. 9 (left and center columns). Coefficients larger (respectively smaller) than 1 mean that the calibrated radiance is increased (respectively decreased) with respect to Version 1. After equalization correction ($C_{eq}(i,c,k)$, left column), the radiances of the flawed rows (green) are increased, with a row-dependent relative amplitude. The same green rows are impacted in the three IIR channels, but the corrections are of different amplitude, as previously discussed. In contrast, the bias corrections ($C_{bias}(i,cycle,k)$, center column) between IIR cycles #51 and #85 decrease the calibrated radiances. They correct the radiometric level of the whole image and are the same for all the rows, but they differ from one channel to another. The Version 2 correction resulting from the product of the equalization and global bias corrections is shown in the right-hand side column of Fig. 9. The weight of the bias correction with respect to the equalization correction increases from IIR1 to IIR2 to IIR3. The relative amplitudes of the corrections for the various rows induce relative changes within an image. The black curves in Fig. 9 are the mean corrections per image, after averaging all the rows. They are useful to picture the changes at large scale, i.e. when one or several images are averaged.

The calibrated radiances (in units of $W.m^{-2}.sr^{-1}.\mu m^{-1}$) are reported in the Level 1b product as integers with a scale factor chosen to report the radiances at a suitable resolution. In Version 1, the scale factor is equal to 100, so that the radiances are reported with a resolution of about 1 % at very cold temperature ($< 200$ K). Because most of the correction coefficients are smaller than 1 % in absolute value, the scale factor has been increased to 1000 in Version 2 to ensure full consistency regardless of the range of radiances.

## 6 Results

The Version 2 calibrated radiances derived after application of the correction coefficients described above are now evaluated.

### 6.1 Striping effect

One of the goals of this work was to reduce the striping effect seen in Version 1 and illustrated in Fig. 1. As seen in Fig. 10, which shows the same cloud-free scene over water surface as in Fig.1, but using Version 2 instead of Version 1, the striping effect is significantly attenuated. The Version 2 IIR inter-channel BTDs along the CALIOP track for the same portion of the same orbit as in Fig. 10 are shown in Fig. 11 for comparison with Version 1 BTDs shown in Fig. 5. The negative peaks which were causing the darker stripes in Version 1 have disappeared and the pixel-to-pixel variability is smaller in Version 2. This indicates that the equalization correction applied in Version 2 has improved the relative calibration of the various rows within an image. In this example, the mean IIR1-IIR3 BTD is increased by 0.4 K, from -1.44 K in Version 1 (Fig. 5) to -1.04 K in





Version 2 (Fig.11), reflecting the different amplitude of the equalization and bias corrections in IIR1 and IIR3. The mean IIR2-IIR3 BTD is increased by only 0.05 K, from 0.89 K in Version 1 to 0.94 K in Version 2, because the corrections applied to these channels are of similar amplitude on average.

## 6.2 Comparisons with MODIS along the CALIOP track

For further evaluation, statistical analyses of IIR-MODIS BTDs along the CALIOP track in clear sky conditions over oceans (Sect. 4.2.2) have been repeated with Version 2 and compared with Version 1. Fig. 12 and Fig. 13 show these comparisons for two different months of two different years, namely July 2008 and January 2013. For both months, the agreement of the median IIR-MODIS BTDs of the sound (purple) and flawed (green) rows is substantially improved in Version 2, owing to the equalization correction. For the sound rows, the difference between Versions 1 and 2 is due to the global bias correction. In
July (Fig. 12), its largest impact, which occurs shortly before cycle #85, is in the descending portion of the orbit until 36° N, where the IIR brightness temperatures are decreased. Thus, the median IIR-MODIS BTDs from the ascending and descending portions of the orbits are in better agreement in Version 2, typically within 0.1-0.2 K.

In January (Fig. 13), the corrections start in the 0-10° N range in the daytime ascending portion of the orbits. The largest corrections occur around the poles where their effects are difficult to assess because of the limited number of samples over
15 water surface and likely contamination by sea ice. Nevertheless, we see by again comparing the sound rows in Versions 1 and Version 2 that the bias correction steadily decreases the Version 2 daytime ascending BTDs, thereby improving the agreement with the nighttime descending BTDs, which are unchanged south of 75° N where IIR cycle numbers are larger than 85.

## 6.3 Stand-alone approach: IIR and MODIS residuals vs. latitude

In G17, the warm bias in Version 1 at night in the 30° N-60° N latitude range could be assessed using the stand-alone approach
by comparing the variations with latitude over oceans of the six IIR and MODIS [simulations-observations] BTDs, called residuals. These comparisons highlighted an unambiguous decrease in nighttime IIR2 and IIR3 residuals from 25° N to 45° N, which was not seen during daytime and was not seen in any of the MODIS residuals. This definitively indicated that nighttime IIR2 and IIR3 residuals were uncharacteristically small in this latitude range, thereby pointing to a warm bias of the nighttime IIR observations.

The experiment conducted with Version 1 is repeated here with Version 2 for the month of July 2008. This analysis is carried out over the IIR swath, with no distinction between the sound and flawed rows. Figure 14 shows the difference between the IIR and MODIS residuals for each pair of IIR-MODIS channels, for IIR Version 1 and Version 2. Each point represents a mean value within a 10-degree latitude range. Between 30° N and 60° N, Version 1 nighttime differences (light blue) are smaller than Version 1 daytime differences (orange) because of the warm bias in IIR nighttime brightness temperatures. The
latter are decreased on average in Version 2 compared to Version 1 between 30° N and 60° N (see Fig. 12), yielding larger nighttime IIR residuals in Version 2 than in Version 1, while the daytime differences remain mostly unchanged. As a result, using IIR Version 2, the agreement between the nighttime (dark blue) and daytime (red) differences is improved between 30°



N and 60° N. The improvement is the most convincing for IIR2 and IIR3, notwithstanding a possible overcorrection by less than 0.2 K at 35° N (30°-40°). IIR1 is the channel for which the night/day differences were the smallest in Version 1 (less than 0.3 K). This agreement is improved at 35° N (30°-40°) in Version 2, but differences smaller than 0.2 K are still present between 40° N and 60° N. North of 60° N, Version 2 daytime IIR2 and IIR3 residuals are slightly larger than in Version 1, by about 0.15 K, while IIR1 ones are mostly unchanged. Overall, the latitudinal variations of the differences between the IIR and MODIS residuals are reduced using IIR Version 2.

## 6.4 Relative approach: nighttime and daytime IIR-MODIS BTDs at 30° N-60° N

As stated in the introduction, this work was motivated in part by unexpected seasonal night/day discrepancies between Version 1 IIR-MODIS BTDs in the 30° N-60° N latitude band observed using the relative approach (Fig. 2). For completeness of this assessment, time series since launch obtained in the same conditions as in Fig. 2 (30° N-60° N and 280-290 K) are shown in Fig. 15 for both Version 1 and Version 2. In the summer, the Version 2 nighttime (dark blue) BTDs are reduced with respect to Version 1 (light blue) while for daytime data, Version 2 (red) and Version 1 (orange) BTDs are mostly identical. The opposite is observed qualitatively for the winter months, but with a weak change in daytime data. As a result, differences between nighttime and daytime BTDs are significantly reduced in Version 2 compared to Version 1. For IIR1-MODIS29, the night/day differences seen in the summer months are reduced by half, from 0.2 K in the worst case in Version 1 to 0.1 K in Version 2. In the winter months, the agreement between day and night BTDs in Version 1 is maintained in Version 2. For IIR2-MODIS31 and IIR3-MODIS32, Version 2 nighttime and daytime BTDs are almost identical. The only exception is IIR3-MODIS32 around June, where nighttime BTDs are smaller than daytime ones by about 0.1 K. It should be noticed that the random noise induced by the instruments is reduced by the averaging process, so that the remaining variability is related to the sensitivity of each IIR-MODIS BTD to surface and atmospheric variability, as discussed in G17.

## 7 Conclusions

Version 2 correction coefficients applied to Version 1 calibrated radiances were defined according to in-depth analyses of the IIR internal calibration coupled with objective quality criteria based on observations, using the same reasoning for the three channels. A two-step correction procedure was defined. The equalization correction corrects the gain of the flawed rows by using sound rows as a reference. The sound rows were rows for which the time variations of the digital counts of the blackbody images, after offset correction, exhibited time variations similar to the radiance of the calibration blackbody source. The rows identified as sound and flawed were found to be the same in the three channels, which is deemed to be a good indication that the same reasoning could be applied for all three channels. Applying the correction between IIR acquisition cycles #46 and #85 for the three channels was found to provide a satisfactory attenuation of the striping effect seen at local scale as well as an agreement at large scale between IIR-MODIS BTDs for the sound and the flawed rows. On the other hand, due to the lack of instrumental evidence, the second global bias correction had to be defined empirically by observing the biases with respect to





MODIS. The global bias correction was defined after determining that the gain of the sound rows -and of the flawed rows after equalization correction, should increase by about 1 between cycles #51 and #85 in the three channels. Version 1 calibration errors were found to steadily increase between cycles #46-51 and #85, as the instrument warms up. They were tentatively explained by the fact that a parasitic signal increasingly affects both the Earth view images and the blackbody images, but differently. The analysis indicates that in the blackbody images, the sound rows are not or only slightly affected, while some parasitic signal is seemingly present in the flawed rows, especially in IIR1. Because of the season-dependent relationship between cycle number and latitude (Fig. 3), these calibration errors induced a hysteresis effect in the IIR-MODIS BTDs in the Northern Hemisphere in the summer months. This hysteresis effect, unambiguously an artifact, explained the warm nighttime biases initially detected using the complementary relative and stand-alone approaches between 30° N and 60° N, mainly in the summer.

It appeared during the course of this study that IIR1 has a specific behavior compared to IIR2 and IIR3. As seen in Fig. 9, the correction coefficients are smaller on average in IIR1 than in IIR2 and IIR3 (black curves), with a larger weight of the equalization correction compared to the global bias correction. In the example for a nighttime descending portion of an orbit on 25 June 2012, between 46° N and 43° N around cycle #80 where the corrections are large, IIR1-IIR3 BTD is increased by 0.4 K on average in Version 2, whereas IIR2-IIR3 BTD is increased by only 0.05 K (Figs. 5 and 10). Both the stand-alone (Sect. 6.3) and the relative (Sect. 6.4) approaches show better corrections in IIR2 and IIR3 than in IIR1 in July between 30° N and 60° N, where nighttime IIR1 seems slightly under-corrected, with remaining differences of less than 0.2 K on average. Attempts to improve the correction in IIR1 in summer while keeping the same reasoning in the three channels were not successful: they were worsening the results in IIR2 and IIR3 or in the winter in IIR1. Notwithstanding this limitation, the improvements in Version 2 with respect to Version 1 are significant. The uncooled micro-bolometer used in the IIR instrument was the first of its kind to be used for radiometric analysis. It appeared to be fully able to meet the radiometric performance required for atmospheric observations.

The IIR Level 1b products have been reprocessed by the CALIPSO project at the NASA Langley Research Center (LaRC) using the calibration corrections described in this paper. The corrections are functions of the IIR cycle number for any orbit of the CALIPSO archive, so that full consistency within the archive is ensured. The new Version 2 of the IIR Level 1b products, which was released in July 2017, is available at the Atmospheric Science Data Center at NASA LaRC and at the AERIS/ICARE Data and Services Center in Lille (France). Version 2 IIR Level 1b products will be used to produce the future Version 4 of the IIR Level 2 products, which is being developed by the IIR science working group in France. Version 4 IIR Level 2 products will also benefit from the improvements in the Version 4 CALIOP products presented in this special issue. The monitoring of the IIR calibrated radiances will continue using the new Version 2 and will be updated using MODIS Collection 6 data. In addition, the stand-alone approach will benefit from the most recent version of the 4A/OP model, a newly released version of the spectroscopic database (GEISA-2015), and will use a clear sky mask derived from Version 4 CALIOP and IIR products.



## Data availability

CALIPSO IIR Level 1b data products are available at the Atmospheric Science Data Center at NASA LaRC (https://eosweb.larc.nasa.gov/project/calipso/calipso_table) and at the AERIS/ICARE Data and Services Center (http://www.icare.univ-lille1.fr). Collocated IIR and MODIS observations from the REMAP product are available at AERIS/ICARE. IIR internal calibration data and post-processed data are available upon request from the authors.

## Competing interests

The authors declare that they have no conflict of interest.

## Acknowledgements

The authors are grateful to NASA LaRC and to SSAI (Science Systems and Applications, Inc.) for their support. This work benefited from the support of the Centre National de la Recherche Scientifique (CNRS) and of Institut National des Sciences de l'Univers (INSU). We thank Brian Getzewich and Tim Murray for the processing of the IIR Level 1b data at NASA LaRC. We thank the AERIS infrastructure for providing access to the CALIPSO and REMAP products, and for data processing. We thank Laurent Crépeau (Laboratoire de Météorologie Dynamique) for generating IIR and MODIS 4A/OP simulations at the AERIS/ESPRI (Ensemble de Services Pour la recherche à l'IPSL) Data and Computing Center of Institut Pierre Simon Laplace (IPSL).

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





**Table 1: IIR acquisition timing for one cycle (see text)**

| 1 Cycle, duration: 40.92 s | | | | | | | | | | | | | | | | | | | | | | | | | | | | | |
|---|---|---|---|---|---|---|---|---|---|---|---|---|---|---|---|---|---|---|---|---|---|---|---|---|---|---|---|---|---|
| Sequence 0 8.184 s | | | | | | Sequence 1 8.184 s | | | | | | Sequence 2 8.184 s | | | | | | Sequence 3 8.184 s | | | | | | Sequence 4 8.184 s | | | | | |
| BB | | | Earth | | | DS | | | Earth | | | DS | | | Earth | | | DS | | | Earth | | | DS | | | Earth | | |
| 1 | 2 | 3 | 3 | 2 | 1 | 1 | 2 | 3 | 3 | 2 | 1 | 1 | 2 | 3 | 3 | 2 | 1 | 1 | 2 | 3 | 3 | 2 | 1 | 1 | 2 | 3 | 3 | 2 | 1 |





**Figure 1: Example of striping effect seen in Version 1 IIR inter-channel BTDs for a cloud-free scene over water surface in the nighttime descending portion of an orbit between 46° N and 43° N on 25 June 2012; (a): CALIOP lidar attenuated backscatter; (b): IIR1-IIR3 BTD; (c): IIR2-IIR3 BTD.**





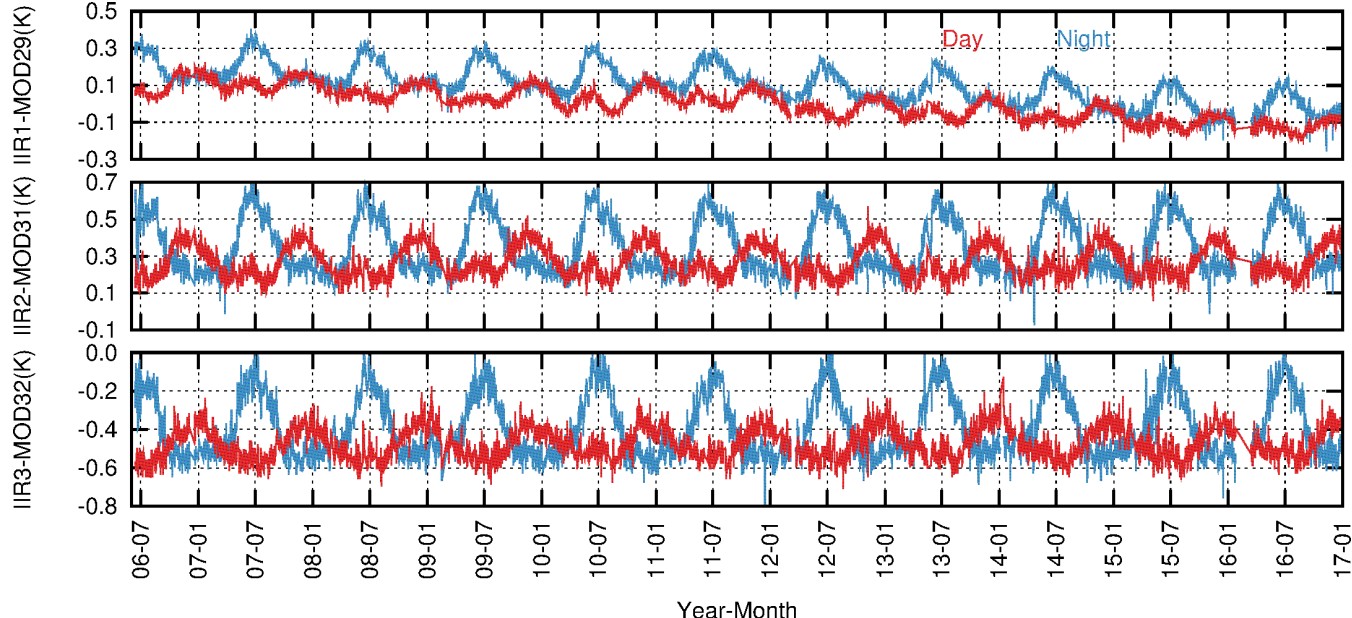

**Figure 2: Time series of IIR Version 1-MODIS C5 daily average BTDs over oceans for IIR1-MODIS29 (top), IIR2-MODIS31 (middle) and IIR3-MODIS32 (bottom). Latitude range: 30° N-60° N. Temperature range: 280-290 K. Unexpected seasonal differences are seen between nighttime (blue) and daytime (red) BTDs. Adapted from Garnier et al. (2017).**





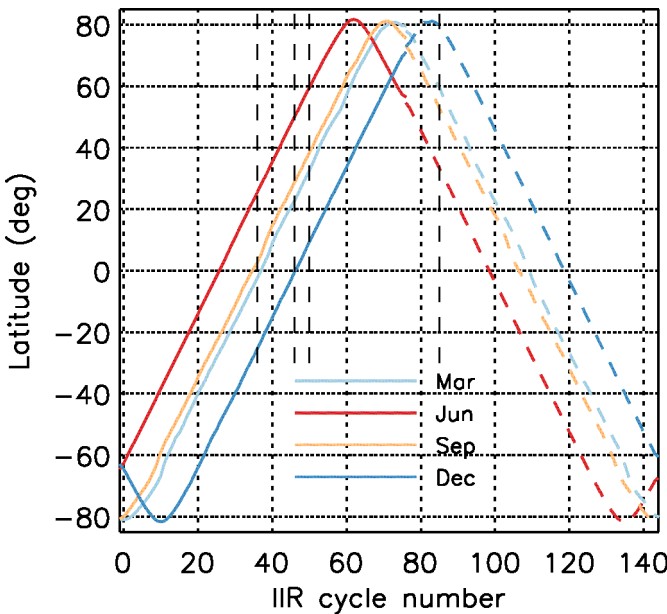

**Figure 3: Relationship between IIR cycle number and latitude in March (light blue), June (red), September (orange), and December (dark blue). Solid: day; dashed: night.**



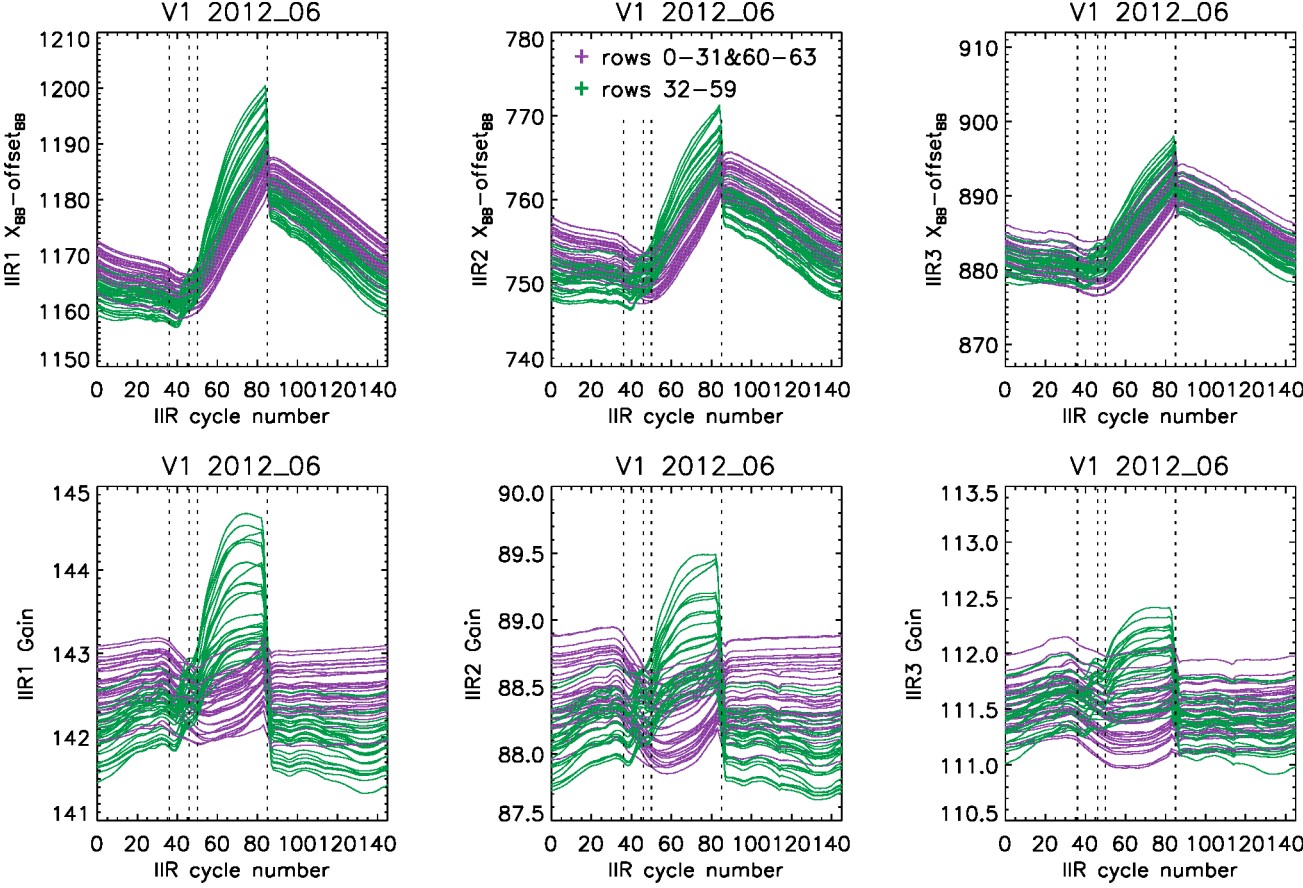

**Figure 4: Mean digital counts in offset-corrected blackbody views (top) and resulting gains (bottom) vs. IIR cycle number in June 2012 for each of the 64 rows, showing a different behavior for the purple (i=0-31 and 60-63) and the green (i=32-59) rows. Digital counts and gain are plotted for the three channels: IIR1 (left), IIR2 (middle), and IIR3 (right).**



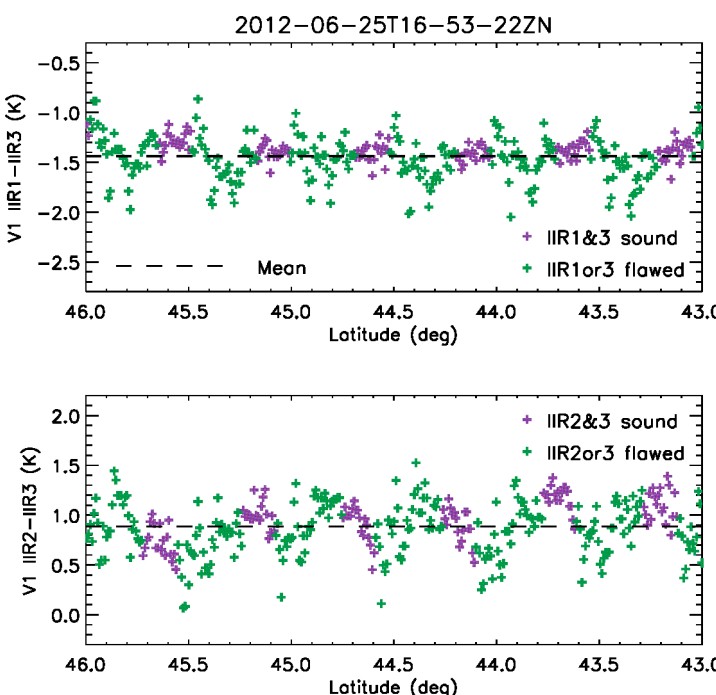

**Figure 5: Version 1 IIR1-IIR3 (top) and IIR2-IIR3 (bottom) inter-channel BTDs along the CALIOP track for the same cloud-free scene over water surface on 25 June 2012 as in Fig. 1. Purple: sound rows in both channels; green: flawed rows in at least one channel. Horizontal dashed line: mean value.**



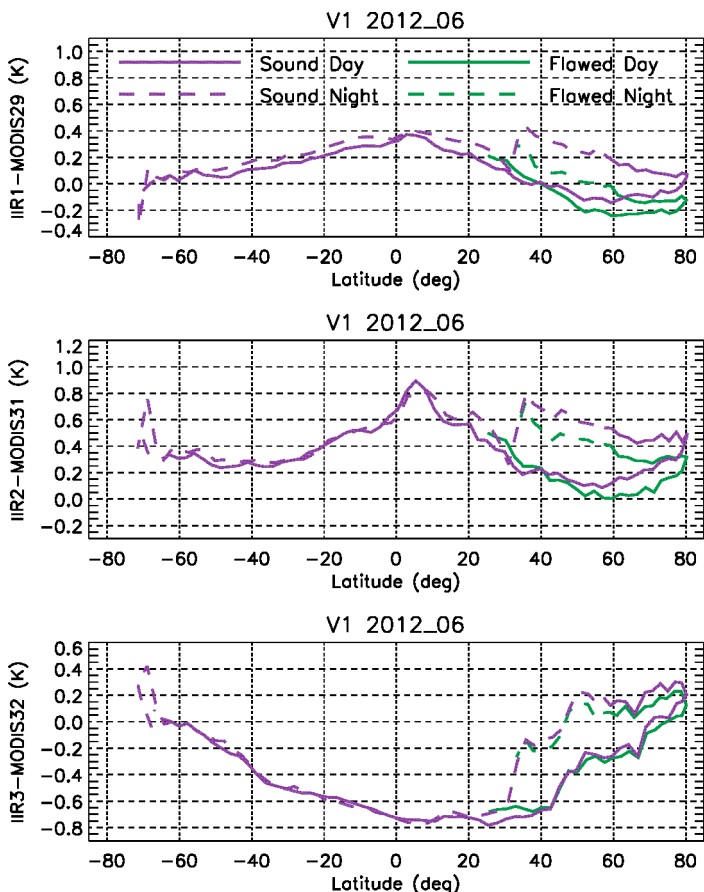

**Figure 6: Median IIR Version 1-MODIS C5 BTDs vs. latitude in clear sky conditions over oceans in June 2012. Top: IIR1-MODIS29; middle: IIR2-MODIS31; bottom: IIR3-MODIS32. Purple: sound rows; green: flawed rows. Solid: day; dashed: night.**



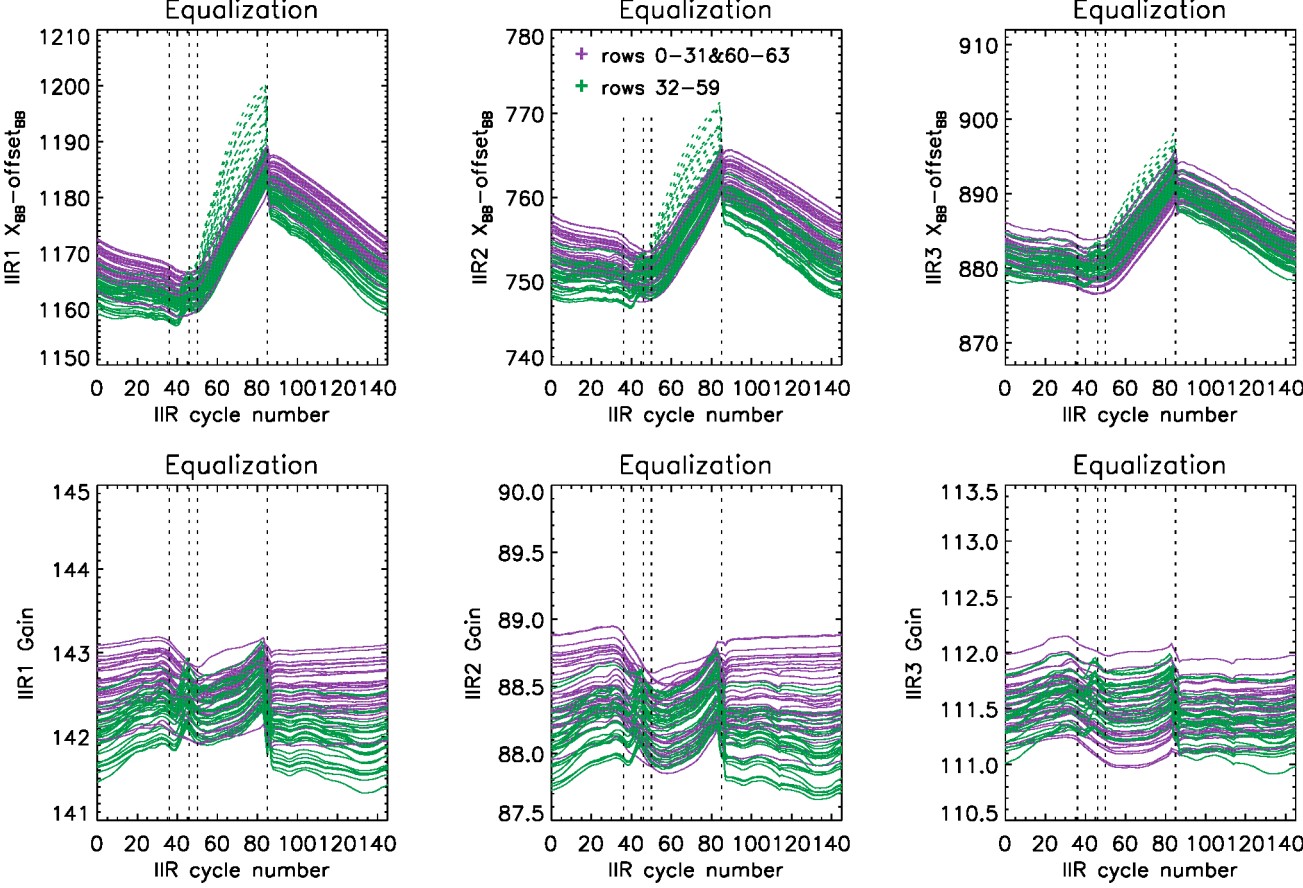

**Figure 7: Mean digital counts in offset-corrected blackbody views after equalization correction (top) and resulting mean gains (bottom) vs. IIR cycle number in June 2012 for each of the 64 rows in IIR1 (left), IIR2 (middle), and IIR3 right). Purple: sound rows; green: flawed rows. The dashed green lines in the upper plots show the flawed rows before correction (cf Fig. 4).**





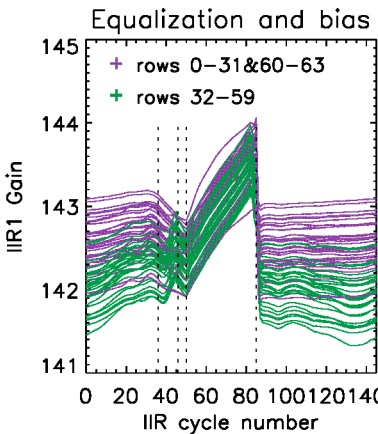 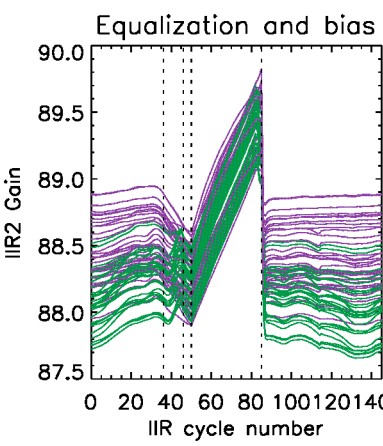 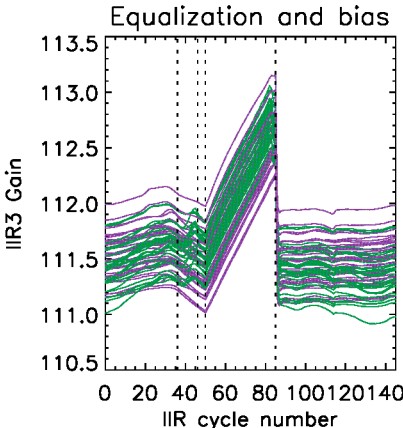

**Figure 8: Mean gains vs. IIR cycle number in June 2012 for each of the 64 rows in IIR1 (left), IIR2 (middle), and IIR3 (right) after equalization and bias corrections. Purple: sound rows; green: flawed rows.**





**Figure 9: Version 2 correction coefficients applied to Version 1 calibrated radiances (top: IIR1, middle: IIR2; bottom: IIR3) vs. IIR cycle number for each of the 64 rows. Left: equalization correction $C_{eq}(i,c,k)$; center: bias correction $C_{bias}(i,c,k)$; right: Version 2 correction, i. e. the product of the equalization and bias corrections. Purple: sound rows; green: flawed rows. The black curve represents the mean correction in an image.**





**Figure 10: Version 2 IIR inter-channel BTDs in the same nighttime descending portion of the same orbit as in Fig. 1. (a): CALIOP lidar attenuated backscatter; (b): IIR1-IIR3 BTD; (c): IIR2-IIR3 BTD. The striping effect is significantly attenuated compared to Version 1.**

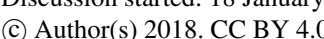




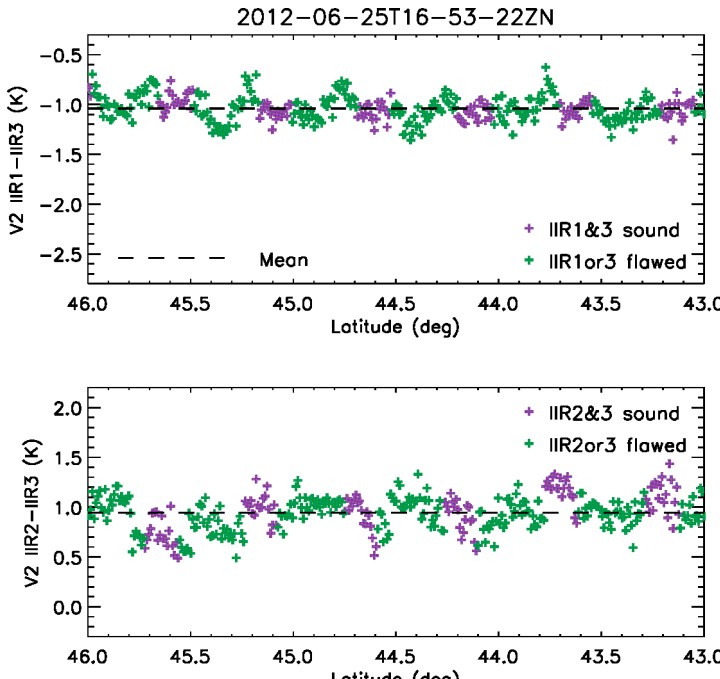

**Figure 11: Version 2 IIR1-IIR3 (top) and IIR2-IIR3 (bottom) inter-channel BTDs in the same nighttime descending portion of the same orbit as in Figs. 1, 5, and 10. Purple: sound rows in both channels; green: flawed rows in at least one channel. Horizontal dashed line: mean value.**





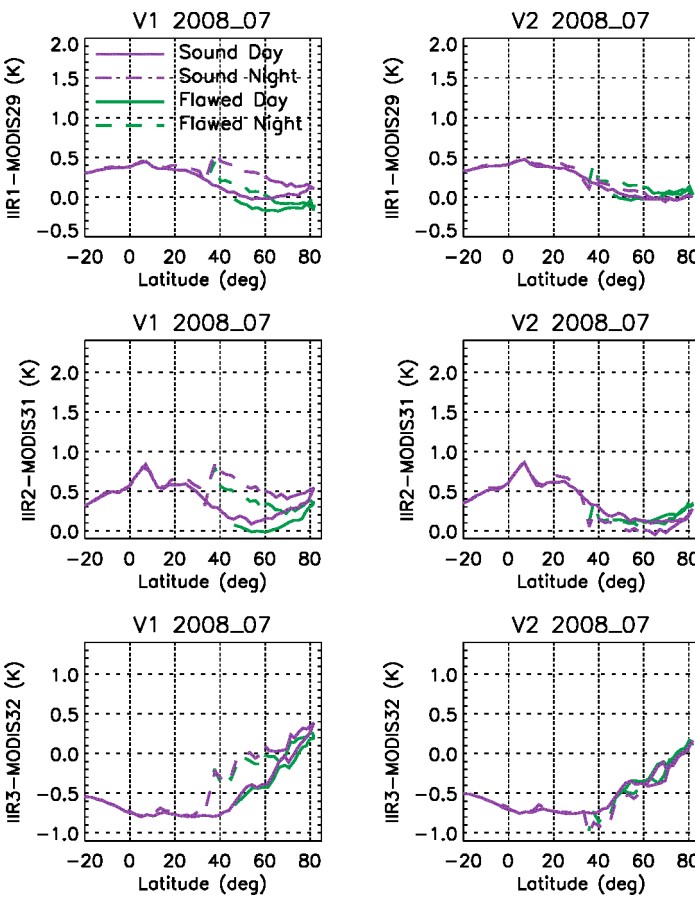

**Figure 12: Median IIR-MODIS C5 BTDs vs. latitude in clear sky conditions over oceans in July 2008 for IIR Version 1 (left) and Version 2 (right). Top: IIR1-MODIS29; middle: IIR2-MODIS31; bottom: IIR3-MODIS32. Purple: sound rows; green: flawed rows. Solid: day; dashed: night.**





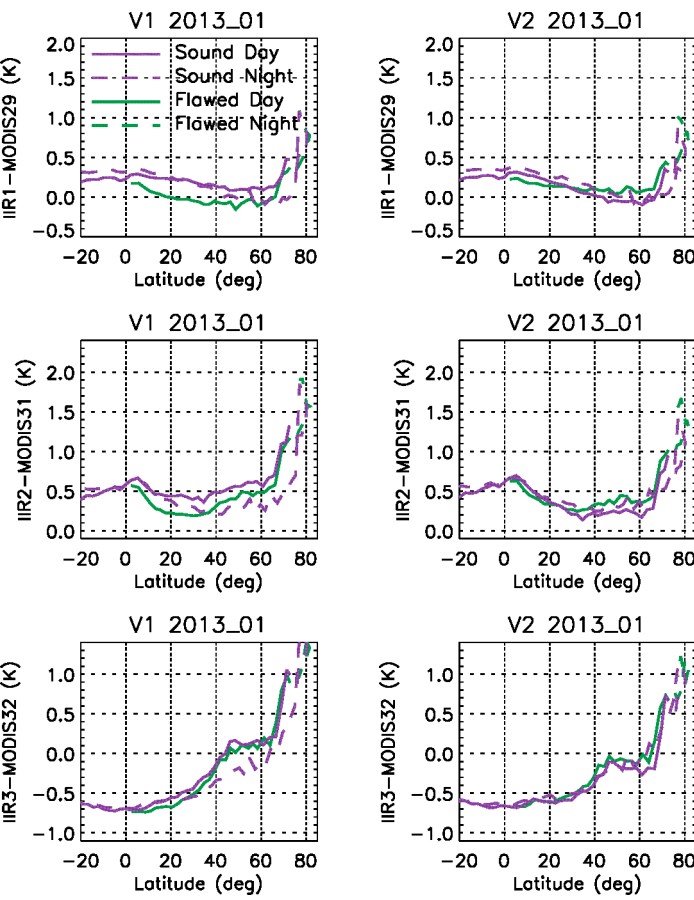

Figure 13: Same as Fig. 12, but for January 2013.





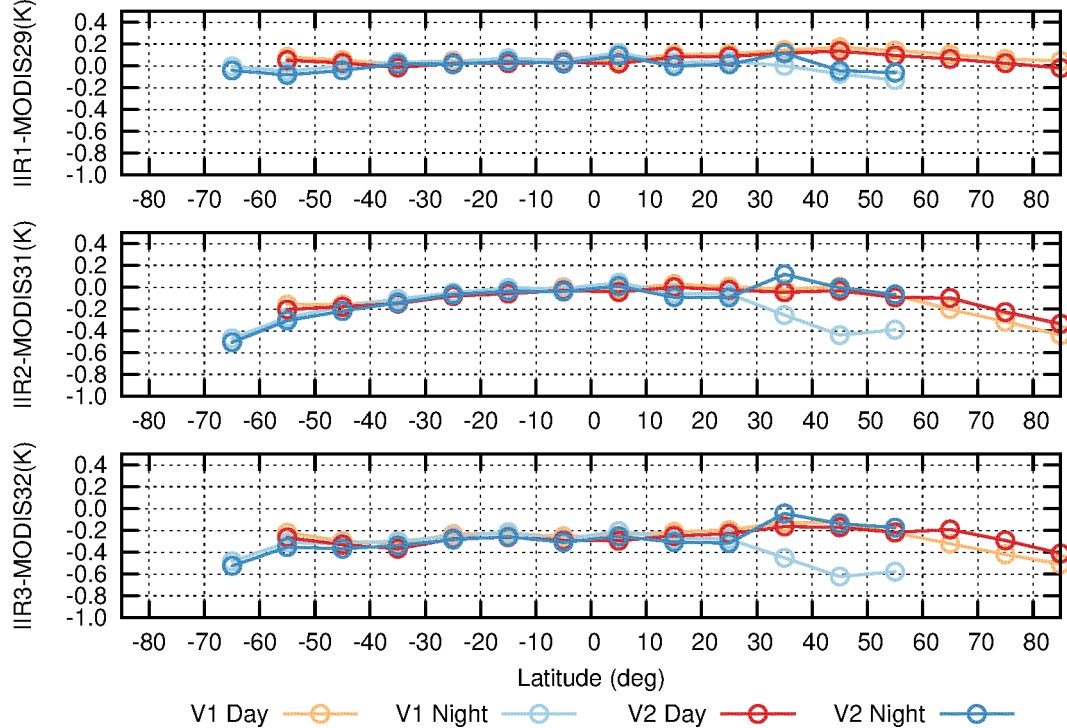

**Figure 14: Difference between IIR and MODIS C5 residuals from the stand-alone approach over oceans for IIR Version 1 (orange: day; light blue: night) and Version 2 (red: day; dark blue: night) in July 2008. Top: IIR1-MODIS29; middle: IIR2-MODIS31; bottom: IIR3-MODIS32.**





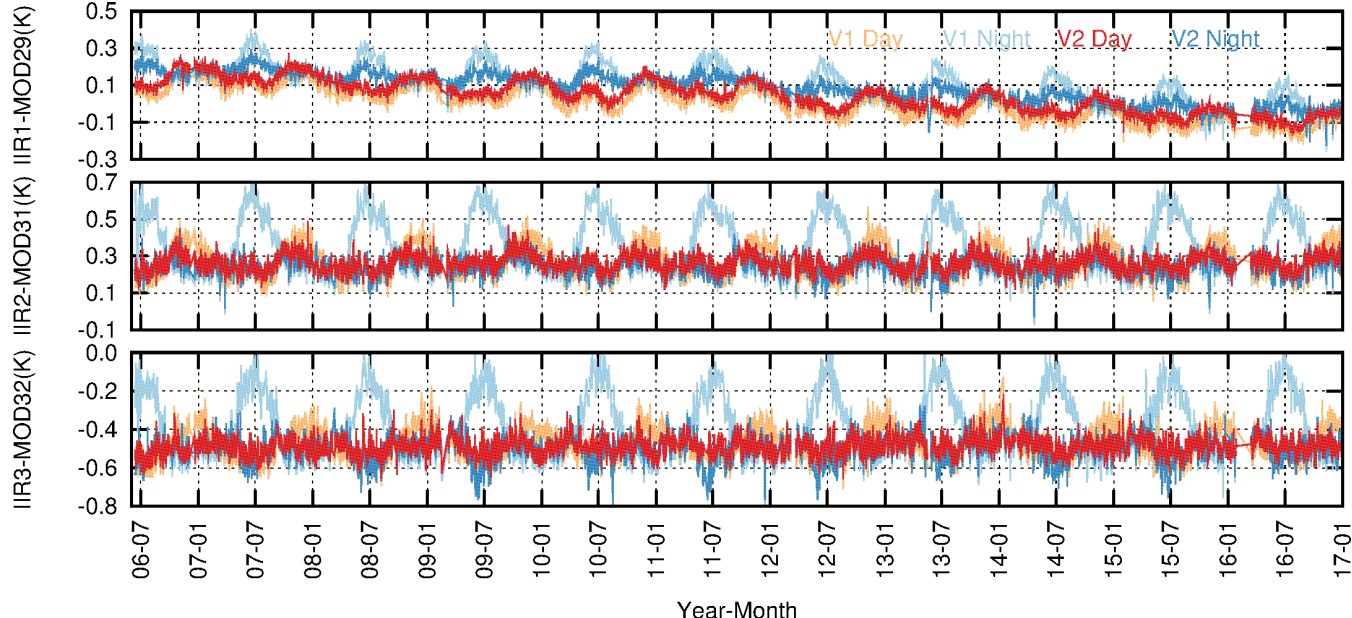

**Figure 15: Time series of IIR-MODIS C5 daily average BTDs over oceans for IIR Version 1 (orange: day; light blue: night) and Version 2 (red: day; dark blue: night). Top: IIR1-MODIS29; middle: IIR2-MODIS31; bottom: IIR3-MODIS32. Latitude range: 30° N-60° N. Temperature range: 280-290 K.**