# Peer review of "CALIPSO IIR Version 2 Level 1b calibrated radiances: analysis and reduction of residual biases in the Northern Hemisphere"

_Atmospheric Measurement Techniques, 2017_

## Referee Comment (RC1) · Anonymous Referee #3 · 11 Feb 2018

This article analyses deficiencies in the calibration of the CALIPSO IIR sensor and proposes an empirical algorithm to mitigate them. The article is clearly presented, based on an exhaustive analysis (albeit within limited range of conditions) and goes into considerable detail. While the authors do not speculate on the underlying cause of the biases found, there are some clues in the results which could be worth further investigation. The benefits of the proposed mitigation algorithm are clearly demonstrated, and will lead to improvements in many applications using these satellite observations.

I only have a few minor corrections and clarifications. Once at least the last two points below are addressed the article would be suitable for publication. The others I would

not consider to be mandatory.

P.2 Line 7 - It would be helpful to mention the equator crossing time.

P.2 Line 9 - How are these bandwidths defined?

P.2 Line 12 - please provide a reference to full details of the definition of equivalent brightness temperature used here.

P.2 Line 20 (and conclusions) - What are the requirements for IIR calibration?

P.3 Line 2 - please add a reference to G17 here.

P.5 Line 18 - does the figure of -0.5K refer to both channel pairs shown?

P.9 Line 2 - This hysteresis effect is interesting. Any idea what could cause it?

P.10 Line 23 - Could the fact that this effect has the same impact on all three channels be a clue to the underlying cause?

P.12 Line 1 - It could be helpful to include values for the standard deviations of the time series shown in Fig. 5 and 11. (The latter could include the former superimposed in feint symbols to highlight the impact.)

P.14 Line 20 - The last two sentences in this paragraph seem out of place here. They warrant a separate paragraph (including a reference to the actual radiometric performance required), and perhaps mention the abstract.

---

## Referee Comment (RC2) · Anonymous Referee #1 · 18 Feb 2018

The paper "CALIPSO IIR Version 2 Level 1b calibrated radiances: analysis and reduction of residual biases in the Northern Hemisphere" presents and discusses the L1b calibrated radiances of the Imaging Infrared Radiometer (IIR) onboard CALIPSO and the improvements of the new Version (Version 2). Two calibration biases revealed in Version 1 initially are addressed: a striping effect of IIR inter-channel BTD and the seasonal warm biases nighttime IIR BT. These technical issues are of critical importance for the quality of the IIR since the biases systematic contaminate the IIR channels. The paper is not only limited to addressing the issue. The paper discusses the developed methodology, the developed semi-empirical approach to deal with the discussed biases and an extended to compare between the two versions, Version 1 and Version 2,

is presented. The study falls within the scope of AMT. The authors have done a thorough job and have a rigorous approach. The manuscript is well-written/structured, the presentation clear, the language fluent and the quality of the figures high. The results support the conclusions. I recommend publication in AMT, however I recommend the following minor revisions before it can proceed to be published.

Comments:

1) Regarding references, a very brief list of references is provided. I would suggest the authors to expand the list of references in order to strengthen the manuscript and at the same in order to give credit to related work. For example in the very first paragraph, at the end of line 7 (page 2) and at line 12 (page 2) suitable references should be made.

2) Page 2, line 8: please provide a more detailed description of the wavelength bandwidths used in IIR1, IIR2 and IIR3.

3) Page 2, line 23: At this point the striping effect is introduced for the first time the manuscript. Although the stripping effect is well established and properly explained and presented, this is done later on in the manuscript, leaving a reader to wonder in the early stages of the manuscript. In that case it would be beneficial for the manuscript to provide at least a brief description of this crucial problem at an earlier stage of the manuscript, maybe through simple referencing to Figure 1.

4) The biases of the IIR are revealed mainly in the geographical domain between 30o N and 60o N. Although the biases, the developed methodology and the improvements are extensively discusses it is not clear the geographical reasons why the IIR channels are contaminated in this domain. I wonder whether the authors can provide an explanation regarding the underlying biases, the causes of the geographical preference in the biases.

5) Page 3, line 1: The authors state that "the analyses revealed that this phenomenon originates from IIR and is due to warm biases in Version 1 nighttime IIR brightness

temperatures in this latitude range". Please provide some more information regarding the analysis and how did the authors reach the conclusion that it is due to the warm biases in V1.

6) Page 4, line 21: The authors state "by averaging digital counts from the eight or nine surrounding DS views". If it is possible provide a more detailed description when and why sometimes the number is 8 and when 9, along with references.

7) Page 8, line 31: The hysteresis effect is very interesting, though it needs further explanation.

8) Page 9, line 22: The authors state that XBB(i)-offsetBB always differ by less that 1.5%. Is the 1.5% a critical value used as boundary limit?

9) Page 9, line 25: What do the authors mean through the term over-correction? Please quantify.

10) Page 10, line 22: What do the authors mean through the term "parasitic contribution"? Please quantify.

11) Page 13, line 5. The authors state that overall, the latitudinal variations of the differences between the IIR and MODIS residuals are reduced using IIR V2. Please quantify.

12) Figure 1a and Figure 10a. The authors should consider to implement the CALIOP official backscatter colormap.

---

## Author Comment (AC1) · 4 Apr 2018

**Response to referee #3**

We are thankful to the reviewer for his/her useful comments that will contribute to greatly improve the manuscript. In the following, the reviewer's comments are in black and our response is in red.

This article analyses deficiencies in the calibration of the CALIPSO IIR sensor and proposes
an empirical algorithm to mitigate them. The article is clearly presented, based
on an exhaustive analysis (albeit within limited range of conditions) and goes into considerable
detail. While the authors do not speculate on the underlying cause of the
biases found, there are some clues in the results which could be worth further investigation.
The benefits of the proposed mitigation algorithm are clearly demonstrated,
and will lead to improvements in many applications using these satellite observations.
I only have a few minor corrections and clarifications. Once at least the last two points
below are addressed the article would be suitable for publication. The others I would
not consider to be mandatory.

P.2 Line 7 - It would be helpful to mention the equator crossing time.

Response

The sentence will be modified as follows:

*"……follows a Sun-synchronous orbit at an altitude of 705 km with **an ascending node equator crossing time at 13:44 local solar time and** an inclination of 98.2°".*

P.2 Line 9 - How are these bandwidths defined?

Response

The IIR spectral response functions will be shown in a new Figure 1 which will be introduced in Sect. 2.1 where the IIR instrument is described. Below is this new figure.

[Figure]

**Figure 1: Spectral response functions in IIR channels IIR1 (black), IIR2 (light grey), and IIR3 (dark grey).**

P.2 Line 12 - please provide a reference to full details of the definition of equivalent brightness temperature used here.

Response

The details will be given in a new Sect. 2.4 and in a new table (Table 2). The new Sect. 2.4 will read:

**2.4 Converting calibrated radiances to brightness temperatures**

*The calibrated radiances reported in the Level 1b product are further converted to brightness temperatures using the Planck's law and the spectral response functions shown in Fig. 1. For each IIR channel, a tabulated function relating radiance (R) in units of $W.m^{-2}.sr^{-1}.\mu m^{-1}$ and equivalent brightness temperature (BT) in units of Kelvin was produced for temperatures ranging between 170 and 330 K. Following a similar approach as developed for previous infrared instruments (e.g., among many others, Weinreb et al. 1997; EUMETSAT, 2012b), we find that for each channel, R can be converted to BT using the equation:*

$$BT = a_0 + (1 + a_1) \cdot BT_{Planck}(R, \lambda_c) \qquad (3)$$

*where $BT_{Planck}(R, \lambda_c)$ is the brightness temperature computed using the Planck's law at wavelength $\lambda_c$, and $a_0$ (in Kelvin) and $a_1$ (unitless) are regression coefficients. The values of $\lambda_c$, $a_0$ and $a_1$ are reported in Table 2 for each IIR channel. Brightness temperatures derived from Eq. (3) and from the tabulated function differ by less than 0.001 K.*

The new Table 2 will be:

*Table 2: Coefficients in Eq. (3) to convert IIR Level 1b radiances (in units of $W.m^{-2}.sr^{-1}.\mu m^{-1}$ ) to equivalent brightness temperatures (in units of Kelvin)*

| IIR channel | $\lambda_c$ (μm) | $a_0$ (K) | $a_1$ (no unit) |
|---|---|---|---|
| IIR1 | 8.621 | -0.768212 | 0.002729 |
| IIR2 | 10.635 | -0.302290 | 0.001314 |
| IIR3 | 12.058 | -0.466275 | 0.002299 |

P.2 Line 20 (and conclusions) - What are the requirements for IIR calibration?

Response

The required calibration accuracy is 1 K for each IIR channel. It is mentioned page 2 line 30, but for more clarity, it will be repeated in Sect 2.1 where we will add :

*"The required calibration accuracy is 1 K for each IIR channel."*

P.3 Line 2 - please add a reference to G17 here.

Response

A reference to G17 will be added and we will also add a referencing to Sect. 3.2. The text will read:

*"Analyses revealed that this phenomenon originates from IIR and is due to warm biases in Version 1 nighttime IIR brightness temperatures in this latitude range (**G17). These analyses are summarized in Sect. 3.2**."*

P.5 Line 18 - does the figure of -0.5K refer to both channel pairs shown?

Response

Yes, the figure of -0.5 K refers to both channels pairs. The text will be modified as follows:

*"In this case, the negative anomaly in the inter-channel BTDs associated to the darker stripes is about -0.5 K **for both pairs of channels**."*

P.9 Line 2 - This hysteresis effect is interesting. Any idea what could cause it?

Response

The following sentence will be added at line 32, page 8:

*"Looking at the relationship between IIR cycle number and latitude in June (Fig. 2), the hysteresis effect indicates that the "global" bias appears after IIR cycle # 40 (35° N in the daytime ascent) and then increases up to cycle # 85 (35° N in the nighttime descent)."*

P.10 Line 23 - Could the fact that this effect has the same impact on all three channels be a clue to the underlying cause?

Response

It could indeed be a clue to the underlying cause, and it could be a good starting point for further analyses of the instrument.

P.12 Line 1 - It could be helpful to include values for the standard deviations of the time series shown in Fig. 5 and 11. (The latter could include the former superimposed in feint symbols to highlight the impact.)

Response

Thank you for this recommendation. These figures (which will be Fig. 6 and 12 in the revised manuscript) have been revised as shown below.

[Figure]

*Figure 6: Version 1 IIR1-IIR3 (top) and IIR2-IIR3 (bottom) inter-channel BTDs along the CALIOP track for the same cloud-free scene over water surface on 25 June 2012 as in Fig. 3. Purple: sound rows in both channels; green: flawed rows in at least one channel. **The horizontal lines show the mean value (dashed) and mean value ± standard deviation (dotted).***

[Figure]

*Figure 12: Version 2 IIR1-IIR3 (top) and IIR2-IIR3 (bottom) inter-channel BTDs in the same nighttime descending portion of the same orbit as in Figs. 3, 6, and 11. Purple: sound rows in both channels; green: flawed rows in at least one channel. **The horizontal lines show the mean value (dashed) and mean value ± standard deviation (dotted) in Version 2 (black) and in Version 1 (grey).***

The text in Sect. 6.1 will be updated accordingly and will read:

*"…….. The Version 2 IIR inter-channel BTDs along the CALIOP track for the same portion of the same orbit as in Fig. 11 are shown in Fig. 12 for comparison with Version 1 BTDs shown in Fig. 6. The negative peaks which were causing the darker stripes in Version 1 have disappeared. **The standard deviation of the inter-channel BTDs is reduced by 40 % from 0.2 K in Version 1 to 0.12 K in Version 2 for the IIR1-IIR3 pair, and by 30 % from 0.26 K to 0.18 K for the IIR2-IIR3 pair. The smaller pixel-to-pixel variability in Version 2** indicates that the equalization correction applied in Version 2 has improved the relative calibration of the various rows within an image …….."*

P.14 Line 20 - The last two sentences in this paragraph seem out of place here. They warrant a separate paragraph (including a reference to the actual radiometric performance required), and perhaps mention the abstract.

Response

We understand the referee's comment.

The first sentence ("*The uncooled micro-bolometer used in the IIR instrument was the first of its kind to be used for radiometric analysis*") is factual information about the instrument. Therefore, we think that it can be moved to Sect. 2.1, which will start as follows:

*"The IIR instrument (Corlay et al., 2000) includes three medium-resolution channels and one unique sensor: an uncooled microbolometer array (U3000A) manufactured by the Boeing company. **The uncooled micro-bolometer used in the IIR instrument was the first of its kind to be used for radiometric analysis**."*

The second sentence is in part justified by previous work (e.g., G17) and seems indeed out of place in the conclusion of this paper. We chose to remove the sentence rather than to develop discussions that would be beyond the scope of this paper.

---

## Author Response (AR1)

**Response to referee #1**

We are thankful to the reviewer for his/her useful comments that will contribute to greatly improve the manuscript. In the following, the reviewer's comments are in black and our response is in red.

The paper "CALIPSO IIR Version 2 Level 1b calibrated radiances: analysis and reduction of residual biases in the Northern Hemisphere" presents and discusses the L1b calibrated radiances of the Imaging Infrared Radiometer (IIR) onboard CALIPSO and the improvements of the new Version (Version 2). Two calibration biases revealed in Version 1 initially are addressed: a striping effect of IIR inter-channel BTD and the seasonal warm biases nighttime IIR BT. These technical issues are of critical importance for the quality of the IIR since the biases systematic contaminate the IIR channels. The paper is not only limited to addressing the issue. The paper discusses the developed methodology, the developed semi-empirical approach to deal with the discussed biases and an extended to compare between the two versions, Version 1 and Version 2, is presented. The study falls within the scope of AMT. The authors have done a thorough job and have a rigorous approach. The manuscript is well-written/structured, the presentation clear, the language fluent and the quality of the figures high. The results support the conclusions. I recommend publication in AMT, however I recommend the following minor revisions before it can proceed to be published.

Comments:

1) Regarding references, a very brief list of references is provided. I would suggest the authors to expand the list of references in order to strengthen the manuscript and at the same in order to give credit to related work. For example in the very first paragraph, at the end of line 7 (page 2) and at line 12 (page 2) suitable references should be made.

Response

At the end of line 7 (page 2) (line 38 page 1 of the marked revised manuscript), we now repeat the reference to Winker et al. (2010). We also added a reference to Stephens et al. (2017) after the reference to Stephens et al. (2012).

At line 12 (page 2) (line 2 page 2 of the marked revised manuscript), we added references to Weinreb et al (1997) and EUMETSAT (2012a). The computation of the equivalent brightness temperatures is now detailed in a new Sect. 2.4 (following a comment by referee #3), where another reference is added: EUMETSAT (2012b).

To summarize, the following references have been added:

EUMETSAT: Effective radiances and brightness temperature relation tables for Meteosat Second Generarion, Rep. EUM/OPS-MSG/TEN/08/0024, 631 pp., Darmstadt, Germany, 2012a.
EUMETSAT: The conversion from effective radiances to equivalent brightness temperatures, Rep. EUM/MET/TEN/11/0569, 49 pp., Darmstadt, Germany, 2012b.
Stephens, G., Winker, D., Pelon, J., Trepte, C., Vane, D., Yuhas, C., L'Ecuyer, T., and Lebsock, M.: CloudSat and CALIPSO within the A-Train: Ten years of actively observing the Earth system, B. Am. Meteorol. Soc., doi:10.1175/BAMS-D-16-0324.1, in press, 2017.
Weinreb, M.P., Jamieson, M., Fulton, N., Chen, Y., Johnson, J.X., Bremer, J., Smith, C., and Baucom, J.: Operational calibration of Geostationary Operational Environmental Satellite-8 and -9 imagers and sounders, *Applied Optics*, 36, 6895-6904, 1997.

2) Page 2, line 8: please provide a more detailed description of the wavelength bandwidths used in IIR1, IIR2 and IIR3.

Response

The IIR spectral response functions are shown in a new Figure 1 which is introduced in Sect. 2.1 where the IIR instrument is described. Below is this new figure:

[Figure]

**Figure 1: Spectral response functions in IIR channels IIR1 (black), IIR2 (light grey), and IIR3 (dark grey).**

3) Page 2, line 23: At this point the striping effect is introduced for the first time the manuscript. Although the stripping effect is well established and properly explained and presented, this is done later on in the manuscript, leaving a reader to wonder in the early stages of the manuscript. In that case it would be beneficial for the manuscript to provide at least a brief description of this crucial problem at an earlier stage of the manuscript, maybe through simple referencing to Figure 1.

Response

We added a brief description and referencing to Sect. 3.1 where the striping effect is presented and illustrated. The text reads as follow (lines 13-15 page 2 of marked revised manuscript):

*"Nevertheless, a striping effect was noticed soon after launch over homogeneous scenes (Trémas, 2006; Scott, 2009). The striping effect* **refers to the presence of stripes in images of IIR inter-channel brightness temperature differences (BTDs) as presented and illustrated in Sect. 3.1***."*

4) The biases of the IIR are revealed mainly in the geographical domain between 30o N and 60o N. Although the biases, the developed methodology and the improvements are extensively discusses it is not clear the geographical reasons why the IIR channels are contaminated in this domain. I wonder whether the authors can provide an explanation regarding the underlying biases, the causes of the geographical preference in the biases.

Response

In the introduction, between lines 24 and 39, page 2 of the marked revised manuscript, we clarified that this study was motivated by the observation of biases only in the Northern Hemisphere and that we are searching for possible sources of biases in the Northern Hemisphere. Thus, the text at lines 27-29, page 2 is now:

*"Both the striping effect and the warm biases in the nighttime IIR calibrated radiances were seen typically* **only** *north of 30° N. These two issues have motivated a detailed examination of the IIR internal calibration procedure and the search for possible sources of biases* **in the Northern Hemisphere**.*"*

In Sect.4, we find calibration biases that are functions of IIR cycle number, which is counted from elapsed time since night-to-day transition. The geographical areas corresponding to the affected IIR cycles result from the season-dependent relationships between IIR cycle number and latitude shown in Fig. 3 of the submitted manuscript (this figure is Fig. 2 in the revised manuscript). This is discussed in Sect. 4.1 (lines 19-26, page 6 of the marked revised manuscript), briefly in Sect. 4.2.1 (line 1, page 7), and more explanations have been added in Sect. 4.2.2 (see comment # 7 about the hysteresis effect).
In Sect. 5.3 about the Version 2 correction coefficients, we added the following sentence (lines 34-35 page 9 of the marked revised manuscript):

*"The Version 2 corrections are between cycles # 46 and #85, in season-dependent portions of the orbits (Fig. 2) that are always located in the Northern Hemisphere."*

In the conclusion, we clarified by modifying the text at lines 1-4 pages 12 of the marked revised manuscript as follows :

*"Because of the season-dependent relationship between cycle number and latitude (Fig. 2), these calibration errors* **were affecting season-dependent latitude ranges always located in the Northern Hemisphere. The calibration errors were detected in the summer months (June/July), because the impacted latitude range was such that they** *induced a hysteresis effect in the IIR-MODIS BTDs in the Northern Hemisphere."*

5) Page 3, line 1: The authors state that "the analyses revealed that this phenomenon originates from IIR and is due to warm biases in Version 1 nighttime IIR brightness temperatures in this latitude range". Please provide some more information regarding the analysis and how did the authors reach the conclusion that it is due to the warm biases in V1.

Response

A reference to G17 has been added as suggested by referee #3, as well as a referencing to Sect. 3.2. The text reads (lines 24-26 page 2 of marked revised manuscript) :

*"Analyses revealed that this phenomenon originates from IIR and is due to warm biases in Version 1 nighttime IIR brightness temperatures in this latitude range (**G17**). **These analyses are summarized in Sect. 3.2**."*

6) Page 4, line 21: The authors state "by averaging digital counts from the eight or nine surrounding DS views". If it is possible provide a more detailed description when and why sometimes the number is 8 and when 9, along with references.

Response

We tried to clarify by changing the end of Sect. 2.3 to :

*"The internal calibration consists in calibrating each pixel of each individual Earth view image by using surrounding DS and BB views (see Table 1). For each channel, and for each pixel in a row (i) and in a column (j) of an individual 64x64 Earth view image in a sequence s, the raw digital counts $X_E(i,j,s)$ are calibrated as follows. First, $X_E(i,j,s)$ is corrected for the offset measured during surrounding DS views. Then, the corrected raw digital counts are converted into calibrated radiances through the gain, $\overline{G}(i,j,s)$. Thus, the calibrated radiance R(i,j,s) in units of $W.m^{-2}.sr^{-1}.\mu m^{-1}$ is written as (**Trémas, 2006**):*

$$R(i, j, s) = \left( X_E(i, j, s) - \text{offset} \right) \times \frac{1}{\overline{G}(i, j, s)} \qquad (1)$$

**The offset and the gain $\overline{G}(i,j,s)$ are derived after averaging several individual DS and BB views, respectively, as was established before launch and confirmed during the in-flight performances assessment (Trémas, 2006). Specifically, the offset is obtained by averaging digital counts from the DS view associated to the sequence, s, if any, and from the eight closest DS views.** *The gain $\overline{G}(i,j,s)$ is obtained by averaging four individual gains associated to the four BB views surrounding the sequence s. An individual gain G(i,j,c) derived from the BB view in a cycle c is computed as:*

$$G(i, j, c) = \frac{X_{BB}(i, j, c) - \text{offset}_{BB}}{R_{BB}(c)} \qquad (2)$$

*where $R_{BB}(c)$ is the blackbody radiance associated with its measured temperature $T_{BB}(c)$, $X_{BB}(i,j,c)$ are the digital counts in the BB view, and $\text{offset}_{BB}$ is the offset correction obtained by averaging the digital counts from the eight closest DS views."*

We noted that Eq. (2) was giving $G^{-1}(i,j,c)$ and not G(i,j,c). We apologize for this mistake which has been corrected in the revised manuscript.

7) Page 8, line 31: The hysteresis effect is very interesting, though it needs further explanation.

Response

The following sentence has been added (lines 32-34 page 7 of marked revised manuscript):

*"Looking at the relationship between IIR cycle number and latitude in June (Fig. 2), the hysteresis effect indicates that the "global" bias appears after IIR cycle # 40 (35° N in the daytime ascent) and then increases up to cycle # 85 (35° N in the nighttime descent)."*

8) Page 9, line 22: The authors state that XBB(i)-offsetBB always differ by less that 1.5%. Is the 1.5% a critical value used as boundary limit?

Response

The value of 1.5 % is not a critical value. We added at the end of the sentence (line 16 page 8 of marked revised manuscript):

*", which was deemed not significant."*

9) Page 9, line 25: What do the authors mean through the term over-correction? Please quantify.

Response

The sentence now reads (lines 18-20 page 8 of marked revised manuscript):

*"Initial attempts to apply the correction between cycles #36 and #85 showed an over-correction **that led to a striping effect as in Version 1, but with anomalous BTDs of opposite sign of about +0.2 K.**"*

10) Page 10, line 22: What do the authors mean through the term "parasitic contribution"? Please quantify.

Response

We tried to clarify by modifying the sentence as (lines 4-8 page 9 of marked revised manuscript):

*The fact that the corrected gains **between cycles #51 and #85** are found to **be larger and to** increase more rapidly than the gains derived after equalization correction (see Fig. 8) suggests **that they correct for** the presence of an additional parasitic contribution to the digital counts in the Earth view images **(see Eq. (1)). This additional contribution represents about 1% of the digital counts in the worst case at cycle #85 in IIR2**.*

11) Page 13, line 5. The authors state that overall, the latitudinal variations of the differences between the IIR and MODIS residuals are reduced using IIR V2. Please quantify.

Response

This statement is indeed difficult to justify and to quantify without a detailed analysis of the remaining longitudinal variations, which is beyond the scope of this study. Therefore, we decided to delete the sentence.

12) Figure 1a and Figure 10a. The authors should consider to implement the CALIOP official backscatter colormap.

Response

These figures (which are Figures 3a and Figure 11a in the revised manuscript) have been modified as suggested. The revised figures are shown below.

[Figure]

**Figure 3: Example of striping effect seen in Version 1 IIR inter-channel BTDs for a cloud-free scene over water surface in the nighttime descending portion of an orbit between 46° N and 43° N on 25 June 2012; (a): CALIOP lidar attenuated backscatter; (b): IIR1-IIR3 BTD; (c): IIR2-IIR3 BTD.**

[Figure]

**Figure 11: Version 2 IIR inter-channel BTDs in the same nighttime descending portion of the same orbit as in Fig. 3. (a): CALIOP lidar attenuated backscatter; (b): IIR1-IIR3 BTD; (c): IIR2-IIR3 BTD. The striping effect is significantly attenuated compared to Version 1.**

**Response to referee #3**

We are thankful to the reviewer for his/her useful comments that will contribute to greatly improve the manuscript. In the following, the reviewer's comments are in black and our response is in red.

This article analyses deficiencies in the calibration of the CALIPSO IIR sensor and proposes an empirical algorithm to mitigate them. The article is clearly presented, based on an exhaustive analysis (albeit within limited range of conditions) and goes into considerable detail. While the authors do not speculate on the underlying cause of the biases found, there are some clues in the results which could be worth further investigation. The benefits of the proposed mitigation algorithm are clearly demonstrated and will lead to improvements in many applications using these satellite observations. I only have a few minor corrections and clarifications. Once at least the last two points below are addressed the article would be suitable for publication. The others I would not consider to be mandatory.

P.2 Line 7 - It would be helpful to mention the equator crossing time.

Response

The sentence has been modified as follows (lines 37-38 page 1 of marked revised manuscript):

*"……follows a Sun-synchronous orbit at an altitude of 705 km with **an ascending node equator crossing time at 13:44 local solar time and** an inclination of 98.2°".*

P.2 Line 9 - How are these bandwidths defined?

Response

The IIR spectral response functions are now shown in a new Figure 1 which is introduced in Sect. 2.1 where the IIR instrument is described. Below is this new figure.

[Figure]

**Figure 1: Spectral response functions in IIR channels IIR1 (black), IIR2 (light grey), and IIR3 (dark grey).**

P.2 Line 12 - please provide a reference to full details of the definition of equivalent brightness temperature used here.

Response

We added "using the Planck's law" (line 2 page 2 of the marked revised manuscript) and the details are given in a new Sect. 2.4 and in a new table (Table 2). The new Sect. 2.4 reads:

*"2.4 Converting calibrated radiances to brightness temperatures*

*The calibrated radiances reported in the Level 1b product are further converted to brightness temperatures using the Planck's law and the spectral response functions shown in Fig. 1. For each IIR channel, a tabulated function relating radiance (R) in units of $W.m^{-2}.sr^{-1}.\mu m^{-1}$ and equivalent brightness temperature (BT) in units of Kelvin was produced for temperatures ranging*

*between 170 and 330 K. Following a similar approach as developed for previous infrared instruments (e.g., among many others, Weinreb et al. 1997; EUMETSAT, 2012b), we find that for each channel, R can be converted to BT using the equation:*

$$BT = a_0 + (1 + a_1) \cdot BT_{Planck}(R, \lambda_c) \qquad (3)$$

*where $BT_{Planck}(R, \lambda_c)$ is the brightness temperature computed using the Planck's law at wavelength $\lambda_c$, and $a_0$ (in Kelvin) and*

5    *$a_1$ (unitless) are regression coefficients. The values of $\lambda_c$, $a_0$ and $a_1$ are reported in Table 2 for each IIR channel. Brightness temperatures derived from Eq. (3) and from the tabulated function differ by less than 0.001 K."*

The new Table 2 is:

10    *Table 2: Coefficients in Eq. (3) to convert IIR Level 1b radiances (in units of $W.m^{-2}.sr^{-1}.\mu m^{-1}$) to equivalent brightness temperatures (in units of Kelvin)*

| IIR channel | $\lambda_c$ (μm) | $a_0$ (K) | $a_1$ (no unit) |
|:---:|:---:|:---:|:---:|
| IIR1 | 8.621 | -0.768212 | 0.002729 |
| IIR2 | 10.635 | -0.302290 | 0.001314 |
| IIR3 | 12.058 | -0.466275 | 0.002299 |

P.2 Line 20 (and conclusions) - What are the requirements for IIR calibration?

Response

The required calibration accuracy is 1 K for each IIR channel. It is mentioned later in the introduction (line 22 page 2 of marked revised manuscript), and for more clarity, it is now repeated in Sect 2.1 where we added (lines 10-11 page 3 of marked revised

20    manuscript) :

*"The required calibration accuracy is 1 K for each IIR channel."*

25    P.3 Line 2 - please add a reference to G17 here.

Response

A reference to G17 has been added and we also added a referencing to Sect. 3.2. The updated text reads (lines 24-26 page 2 of

30    the marked revised manuscript) :

*"Analyses revealed that this phenomenon originates from IIR and is due to warm biases in Version 1 nighttime IIR brightness temperatures in this latitude range (**G17**). **These analyses are summarized in Sect. 3.2**."*

35    P.5 Line 18 - does the figure of -0.5K refer to both channel pairs shown?

Response

Yes, the figure of -0.5 K refers to both channels pairs. The text has been modified as follows (line 3 page 5 of marked revised

40    manuscript):

*"In this case, the negative anomaly in the inter-channel BTDs associated to the darker stripes is about -0.5 K **for both pairs of channels**."*

45

P.9 Line 2 - This hysteresis effect is interesting. Any idea what could cause it?

Response

50    The following sentence has been added (lines 32-34 page 7 of marked revised manuscript):

*"Looking at the relationship between IIR cycle number and latitude in June (Fig. 2), the hysteresis effect indicates that the "global" bias appears after IIR cycle # 40 (35° N in the daytime ascent) and then increases up to cycle # 85 (35° N in the nighttime descent)."*

P.10 Line 23 - Could the fact that this effect has the same impact on all three channels be a clue to the underlying cause?

Response

10 Yes, it could be a clue to the underlying cause. This effect is seen when the temperature of the instrument increases, suggesting a thermally induced effect. No straightforward information about the underlying cause could be found in the available instrument data so far. We agree that the conclusions from this study could be a good starting point for further analyses of the IIR instrument.

15 P.12 Line 1 - It could be helpful to include values for the standard deviations of the time series shown in Fig. 5 and 11. (The latter could include the former superimposed in feint symbols to highlight the impact.)

Response

20 Thank you for this recommendation. These figures (which are Fig. 6 and 12 in the revised manuscript) have been revised as shown below.

[Figure]

[Figure]

*Figure 6: Version 1 IIR1-IIR3 (top) and IIR2-IIR3 (bottom) inter-channel BTDs along the CALIOP track for the same cloud-free scene over*
25 *water surface on 25 June 2012 as in Fig. 3. Purple: sound rows in both channels; green: flawed rows in at least one channel. **The horizontal lines show the mean value (dashed) and mean value ± standard deviation (dotted).***

[Figure]

[Figure]

*Figure 12: Version 2 IIR1-IIR3 (top) and IIR2-IIR3 (bottom) inter-channel BTDs in the same nighttime descending portion of the same orbit as in Figs. 3, 6, and 11. Purple: sound rows in both channels; green: flawed rows in at least one channel.* ***The horizontal lines show the mean value (dashed) and mean value ± standard deviation (dotted) in Version 2 (black) and in Version 1 (grey).***

The text in Sect. 6.1 has been updated accordingly and reads (lines 6-11 page 10 of marked revised manuscript):

*"…….. The Version 2 IIR inter-channel BTDs along the CALIOP track for the same portion of the same orbit as in Fig. 11 are shown in Fig. 12 for comparison with Version 1 BTDs shown in Fig. 6. The negative peaks which were causing the darker stripes in Version 1 have disappeared.* ***The standard deviation of the inter-channel BTDs is reduced by 40 % from 0.2 K in Version 1 to 0.12 K in Version 2 for the IIR1-IIR3 pair, and by 30 % from 0.26 K to 0.18 K for the IIR2-IIR3 pair. The smaller pixel-to-pixel variability in Version 2*** *indicates that the equalization correction applied in Version 2 has improved the relative calibration of the various rows within an image…….."*

P.14 Line 20 - The last two sentences in this paragraph seem out of place here. They warrant a separate paragraph (including a reference to the actual radiometric performance required), and perhaps mention the abstract.

Response

We understand the referee's comment.

The first sentence (*"The uncooled micro-bolometer used in the IIR instrument was the first of its kind to be used for radiometric analysis"*) is factual information about the instrument. Therefore, we think that it can be moved to Sect. 2.1, which now starts as follows:

*"The IIR instrument (Corlay et al., 2000) includes three medium-resolution channels and one unique sensor: an uncooled microbolometer array (U3000A) manufactured by the Boeing company.* ***The uncooled micro-bolometer used in the IIR instrument was the first of its kind to be used for radiometric analysis.***"

The second sentence is in part justified by previous work (e.g., G17) and seems indeed out of place in the conclusion of this paper. We chose to remove the sentence rather than to develop discussions that would be beyond the scope of this paper.

[revised manuscript text omitted]